# Hominoid-specific transposable elements reshaped neural crest migration in craniofacial development

Laura Deelen (ID), Zoe H Mitchell (ID), Martina Demurtas, Andria Koulle, Beatriz Garcia Del Valle & Marco Trizzino (ID) ✉

## Abstract

**Craniofacial development is evolutionarily conserved, yet subtle changes in its regulatory network drive species-specific traits. Transposable elements (TEs) contribute to genome evolution, but their role in cranial neural crest cells (CNCCs) remains unclear. Here, we investigate the domestication of hominoid-specific TEs (LTR5Hs and SVAs) as enhancers during human CNCC specification, a process critical for vertebrate craniofacial development. Using human iPSC-derived CNCCs, we identified ~515 hominoid-specific TEs functioning as enhancers, including ~250 human-specific, predominantly LTR5Hs. These elements are enriched for CNCC coordinator motifs, are bound by the CNCC signature factor TWIST1, and their enhancer activity appears largely CNCC-specific. CRISPR-interference targeting ~75% of these active TEs led to widespread transcriptional dysregulation of genes involved in neural crest migration, and two orthogonal functional assays confirmed that CNCC migration is impaired upon TE repression. Finally, genes near human-specific TEs showed higher expression in human CNCCs compared to chimpanzee CNCCs, but TE repression restored gene expression to chimpanzee levels. These findings highlight how young TEs were domesticated to fine-tune CNCC regulatory networks, potentially contributing to lineage-specific craniofacial evolution.**

**Keywords** Cell Migration; Coordinator Motif; Cranial Neural Crest; LTR5Hs; SVAs
**Subject Categories** Evolution & Ecology; Genetics, Gene Therapy & Genetic Disease

## Introduction

Craniofacial development is a highly complex process that requires precise spatiotemporal regulation and involves contributions from all three germ layers, and in particular from neural crest cells (NCCs) (Gans and Northcutt, 1983; Ahlstrom and Erickson, 2009; Martik and Bronner, 2021; Theveneau and Mayor, 2012). NCCs emerge during early embryogenesis, between weeks 3–4 in humans, at the neural plate border between the neuroectoderm and non-neural ectoderm. As neurulation progresses, the neural plate invaginates and separates from the dorsal ectoderm, forming the neural tube. At this stage, NCCs undergo epithelial-to-mesenchymal transition (EMT), allowing them to delaminate and migrate to specific regions throughout the developing embryo (Ahlstrom and Erickson, 2009; Martik and Bronner, 2021; Theveneau and Mayor, 2012). Among the NCC subtypes, cranial neural crest cells (CNCCs) play a pivotal role in the formation of key craniofacial structures, including bones and cartilage (Bronner and LeDouarin, 2012; Cordero et al, 2011; Jheon and Schneider, 2009).

Although craniofacial development is an evolutionarily conserved process, recent adaptations to the modern human craniofacial complex include changes in shape and function to accommodate the enlargement of the brain, the transition to bipedal posture, laryngeal extension for speech, as well as adjustments for the evolvement of sensory organs (Lieberman, 1998; Sambataro et al, 2022; Spoor et al, 1994). The evolution of human-specific craniofacial traits has required precise modifications in gene expression, and increasing evidence suggests that regulatory changes, rather than protein-coding mutations, have been key to shaping species-specific features (Carroll, 2005; King and Wilson, 1975; Wray, 2007). One major source of these regulatory innovations are transposable elements (TEs). Comprising nearly half of the human genome, TEs are now recognized as key contributors to genomic evolution through their ability to integrate into the genome and act as cis-regulatory elements (Bourque et al, 2008; Chuong et al, 2013, 2016; Cosby et al, 2021; Goubert et al, 2020; Kunarso et al, 2010; Lynch et al, 2011; Pontis et al, 2019; Schmidt et al, 2012; Sundaram and Wysocka, 2020). TE-mediated rewiring of gene regulatory networks has previously been implicated as a major driver of species-specific gene expression patterns (Chuong et al, 2013; Feschotte, 2008; Fueyo et al, 2022; Jacques et al, 2013; Patoori et al, 2022; Playfoot et al, 2021; Prescott et al, 2015; Sundaram et al, 2014; Trizzino et al, 2017).

However, much remains to be uncovered about the precise mechanisms through which TEs have been co-opted as cis-regulatory elements in humans, particularly in the context of craniofacial development. In the human genome, LTR5Hs and SINE-VNTR-Alus (SVAs) have been previously linked to gene regulatory activity (Barnada et al, 2022; Chuong et al, 2016; Fuentes et al, 2018; Preprint: Fueyo et al, 2025; Patoori et al, 2022; Pontis et al, 2019; Trizzino et al, 2017).

Department of Life Sciences, Imperial College London, SW7 2AZ London, UK. ✉E-mail: m.trizzino@imperial.ac.uk

LTR5Hs represent the youngest class of long terminal repeats (LTRs), which flank the most recently expanded human endogenous retroviruses (ERVs), HERV-K (HML2) (Fig. 1A). Due to their flanking positions at proviral sequences, LTRs have evolutionarily functioned as retroviral promoters or other regulatory elements. However, the vast majority of LTR5Hs copies now reside in the genome as 'solo' LTRs as a result of homologous recombination between the flanking LTRs (Subramanian et al, 2011). There are currently around 700 annotated copies in the human genome (GRCh38).

SVAs are the youngest TE family, and over 6000 copies are currently annotated in the human genome (GRCh38). They are composed of a hexamer repeat, an Alu-like element, a variable number of tandem repeats (VNTRs), a SINE element, and a poly-A tail (Fig. 1A). The SVAs are hominoid-specific, comprise six subfamilies (A–F), with SVA-E and SVA-F being found exclusively in humans, with ~1700 copies in total (Quinn and Bubb, 2014; Wang et al, 2005).

Co-option of LTR5Hs and SVAs as active cis-regulatory elements has been observed in different human tissues (Barnada et al, 2022; Chuong et al, 2013; Fuentes et al, 2018; Ostertag et al, 2003; Patoori et al, 2022; Pontis et al, 2019; Trizzino et al, 2017). Since the expansion of LTR5Hs and SVA-E/F subfamilies occurred around the time of the human–chimpanzee split, we set out to investigate if these elements could have contributed to human-specific craniofacial development.

Human cranial neural crest cell (CNCC) specification and migration can be effectively modeled in vitro using human-induced pluripotent stem cells (hiPSCs). Therefore, in this study, we employed an inducible CRISPR-interference (CRISPRi) hiPSC line to investigate the impact of silencing human LTR5Hs and SVA elements on CNCC specification and migration, which are essential processes for the development of the craniofacial structures, including bones and cartilage. To achieve this, we employed previously published single-guide RNAs (sgRNAs) targeting approximately 80% of these transposable elements (Pontis et al, 2019).

With this approach, we identified ~515 hominoid-specific TEs (predominantly LTR5Hs) that are accessible and depleted of the repressive histone mark H3K9me3 in human CNCCs. We found that the specific DNA sequence was the primary driver for the co-option of this set of TEs. Importantly, silencing these retroelements attenuated the expression of hundreds of genes involved in CNCC migration, which is a key process in craniofacial morphogenesis. Functional assays confirmed that migration was disrupted, suggesting a potential role for the transposons in species-specific craniofacial development. This was further supported by comparisons with previously published chimpanzee CNCC data, where the corresponding genes to those near accessible human-specific transposable elements showed expression patterns similar to those in human CNCCs when these elements were silenced.

## Results

### An inducible CRISPR-interference iPSC system for the repression of LTR5Hs and SVAs

To investigate the role of hominoid-specific TEs (LTR5Hs and SVAs; Fig. 1A) in human cranial neural crest cell (CNCC) development, we designed a stable human iPSC line with an inducible CRISPR-interference (CRISPRi) system with single-guide

RNAs (sgRNAs) targeting ~80% of all LTR5Hs and SVAs annotated in the human genome (Fig. 1B). The sgRNAs were originally designed and validated in a recent seminal study (Pontis et al, 2019) and were further used in subsequent studies (Barnada et al, 2022; Patoori et al, 2022). Briefly, we cloned a stable iPSC line with a permanently integrated TET-inducible, catalytically dead, Cas9 fused to a repressive KRAB domain (dCas9-KRAB), along with the two gRNAs targeting LTR5Hs and SVAs (hereafter +gRNA line). The KRAB domain recruits the transcriptional machinery necessary to deposit repressive histone methylation (H3K9me3) to the regions targeted by dCas9. To account for potential off-target effects caused by exposure to doxycycline (TET-ON) or by Cas9 expression, we cloned the same iPSC line with an identical dCas9-KRAB construct but without any gRNAs (hereafter −gRNAs line). Treating the cells with doxycycline for 24 h was sufficient to activate dCas9 in both CRISPRi lines (i.e., with and without gRNAs; Fig. 1C,D).

Next, we generated CNCCs from our CRISPRi-iPSC lines using an established 5-day protocol (Fig. 1E,F; Leung et al, 2016). Since both LTR5Hs and SVAs have been shown to have important roles in human embryonic stem cells and iPSCs (Barnada et al, 2022; Fuentes et al, 2018; Pontis et al, 2019), doxycycline was only introduced 24 h after differentiation (Fig. 1C). By day 5 of differentiation, both cell lines expressed markers typical of CNCC identity, both at gene (SOX9, SOX10, TWIST1, and TFAP2A) and protein (SOX9 and AP2α) level. We generally did not observe significant differences in CNCC marker expression between the two lines, apart from TFAP2A (Fig. 1E,F), indicating that expression of the main genes essential for CNCC identity was largely unaffected by the CRISPRi.

Finally, using RNA-seq data, we explored the TE expression landscape in human CNCCs. This revealed that many TE subfamilies are highly expressed (including LTR5Hs (ERVK:LTR) and SVAs), with Alu (SINE) and L1 (LINE) elements dominating (Appendix Fig. S1A; Dataset EV1).

### Hundreds of LTR5Hs and SVAs are accessible in human CNCCs

We first set out to determine whether any LTR5Hs and SVAs exhibited chromatin accessibility in human CNCCs, and whether these elements could be repressed using our CRISPRi system. To this end, we performed ATAC-seq (paired-end, 150 bp reads) in hiPSC-derived CNCCs generated with our CRISPRi-iPSC lines (− and +gRNAs). We started by examining human-specific SVAs (i.e., SVA-E and SVA-F) and LTR5Hs. K-mer clustering of the ATAC-seq data identified a total of 256 accessible LTR5Hs and human-specific SVAs (Fig. 2A; Dataset EV2). Notably, 77% of these 256 TEs displayed decreased accessibility in the +gRNA CRISPRi line relative to the −gRNA control (cluster 2, Fig. 2A). This is consistent with the assumption that these gRNAs can target ~80% of all the human LTR5Hs and SVAs (Pontis et al, 2019). LTR5Hs were significantly overrepresented among the 256 accessible human-specific TEs (observed 87%, expected 29%; Fisher's Exact Test $p < 2.2 \times 10^{-16}$; Fig. 2B), while the SVAs were significantly underrepresented (13% observed vs 71% expected; Fisher's exact test $p < 2.2 \times 10^{-16}$; Fig. 2B), highlighting a potential primary role for LTR5Hs in CNCCs.

We set out to confirm that the loss in chromatin accessibility observed in the +gRNA lines was a direct consequence of the

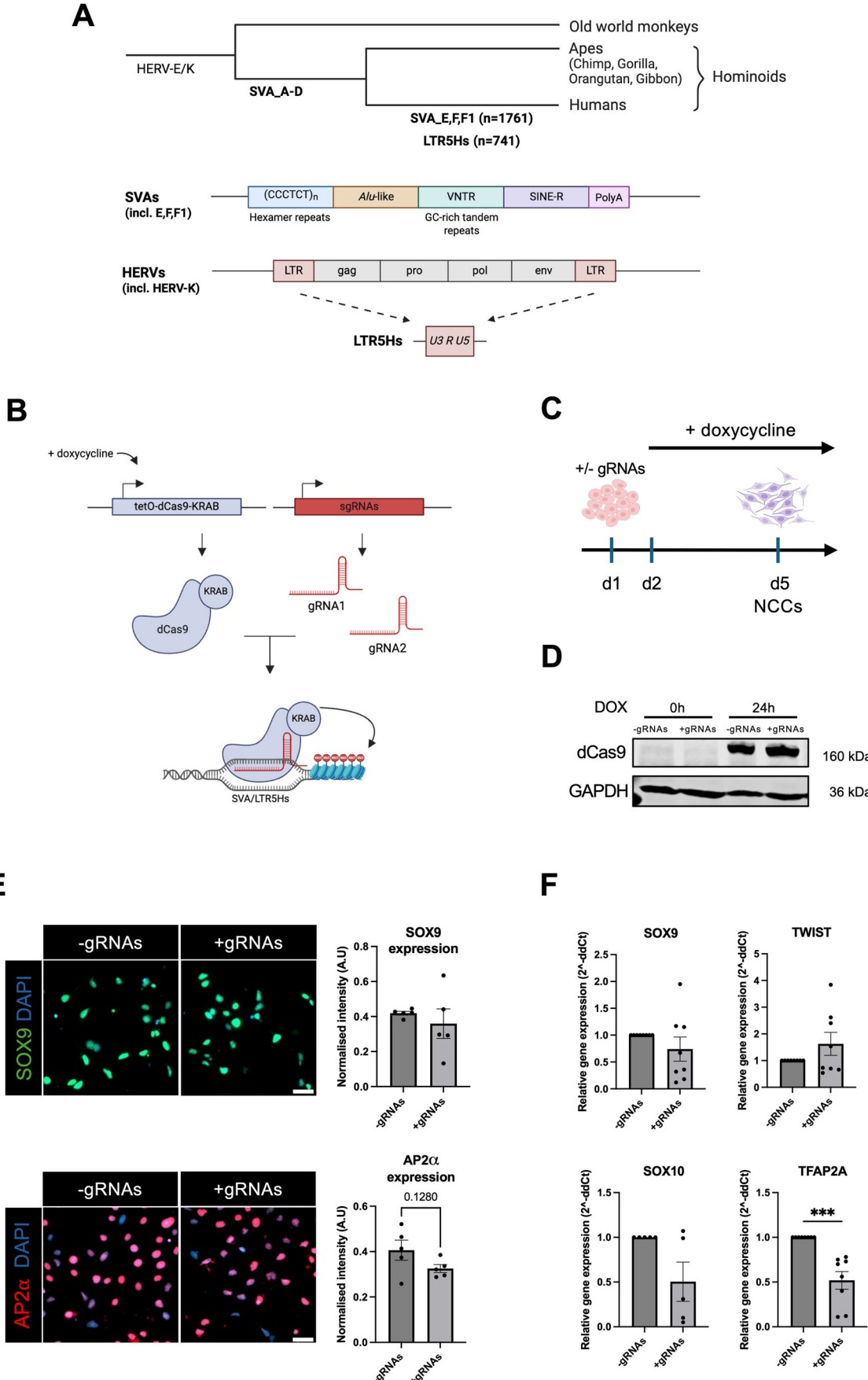

**Figure 1. An inducible CRISPRi tool to repress human-specific SVAs and LR5Hs in iPSC-derived CNCCs.**

(A) Schematic overview of SVA and LTR5Hs transposable elements, illustrating their structure and phylogenetic distribution across Old World monkeys, apes, and humans. Human-specific SVA subfamilies (i.e., SVA_E/F/F1) and the LTR5Hs are highlighted, along with the approximate number of elements in each category. (B) CRISPR-dCas9-KRAB targeting strategy for SVAs and LTR5Hs. Two guide RNAs (gRNA1 and gRNA2) target ~80% of all SVAs and LTR5Hs, directing dCas9-KRAB to deposit repressive epigenetic marks at the TEs, thereby reducing their transcriptional activity. (C) Experimental timeline for doxycycline (dox)-induced expression of dCas9-KRAB. Dox is added to the media 24 h after the start of differentiation into cranial neural crest cells, which spans 5 days. (D) Western blot confirming dox-inducible dCas9-KRAB expression. Lysates from cells cultured with or without TE-targeting gRNAs were probed with antibodies against dCas9 (top band, ~160 kDa) and GAPDH (bottom band, loading control). (E) Immunofluorescence displaying expression of CNCC signature markers AP2α (red) and SOX9 (green) in CNCCs differentiated using CRISPRi-iPSCs without (left) and with gRNAs (right). Signal intensity was quantified and normalized to the cell number per frame. This experiment was repeated at least once. Scale bar: 50 μm. (F) RT-qPCR of CNCC markers SOX9, TWIST1, SOX10 and TFAP2A. n = 6: each data point is derived from six independent rounds of NCC differentiation. Statistical analyses were performed using unpaired t-tests. Error bars indicate mean ± SEM. ***p = 0.0002. Source data are available online for this figure.

CRISPRi-mediated repression. To this end, we performed ChIP-seq for H3K9me3 (paired-end 150 bp reads), which showed accumulation of this repressive histone mark at ~75% of the 256 accessible LTR5Hs and human-specific SVAs in the +gRNA samples (Fig. 2C). Notably, there was a remarkable overlap (92%) between the TEs that lost chromatin accessibility (Fig. 2A) and those gaining H3K9me3 (Fig. 2C).

We further investigated the genomic locations of these accessible human-specific TEs. Overall, 91% of the 256 accessible LTR5Hs and human-specific SVAs were located >1 kb from the nearest transcription start site (TSS), and the median distance was 8.9 kb (Appendix Fig. S1B). This suggests that these accessible mobile elements could be putative CNCC enhancers. To confirm this, we leveraged publicly available H3K27ac ChIP-seq data previously generated by our group in hiPSC-derived CNCCs (Barnada et al, 2024; Data ref: Barnada et al, 2024). This analysis revealed that the vast majority of the 256 accessible human-specific TEs also displayed the H3K27ac modification, an established active enhancer mark, supporting their role as bona-fide human-specific CNCC enhancers (Appendix Fig. S1C).

To investigate whether these accessible human-specific TEs are actively transcribed in CNCCs, we performed an H3K4me3 ChIP-seq (which is a marker for actively transcribing promoters) in −gRNA CNCCs and +gRNA CNCCs. This experiment revealed that 39.5% of the accessible LTR5Hs and human-specific SVAs were also decorated by H3K4me3, suggesting that over a third of the accessible LTR5Hs and human-specific SVAs are likely engaged in active transcription in CNCCs (Appendix Fig. S1D,E).

Next, we focused on the accessibility of all SVA subfamilies, including those that are also found in other hominoid species (SVA_A, _B, _C, _D). To this end, we analysed the ATAC-seq signal across all existing SVA subfamilies and LTR5Hs separately (Appendix Fig. S2A,B). This yielded results consistent with the previous analysis, since we found that 5% (n = 258) of non-human-specific SVAs (A–D subfamilies) were accessible in CNCCs, whilst 95% (n = 5716) were inaccessible (Appendix Fig. S2A,B). Therefore, even when accounting for all SVA groups, SVAs remained significantly underrepresented among accessible TEs.

In summary, these experiments identified approximately 515 hominoid-specific TEs, over half of which are human-specific, that exhibit signatures of active enhancers in human iPSC-derived CNCCs, with LTR5Hs elements playing a prominent role. Notably, our CRISPR-based approach enabled the inducible repression of ~75% of these putative TE-derived human-specific CNCC enhancers.

To ensure that our findings were not biased by a specific differentiation protocol, we used an alternative iPSC-to-CNCC differentiation method (Bajpai et al, 2010) and performed ATAC-seq on the differentiated cells. With this protocol, migratory CNCCs are obtained in 2–3 weeks (Bajpai et al, 2010; Barnada et al, 2024; Mitchell et al, 2025; Pagliaroli et al, 2021). Using this approach, we identified 374 LTR5Hs and human-specific SVA elements that are accessible in CNCCs, the majority of which exhibited reduced accessibility in the +gRNA line compared to the −gRNA control (Appendix Fig. S2C). Moreover, comparable to our findings obtained with the 5-day protocol, LTR5Hs were significantly overrepresented in the set of accessible TEs (83% observed vs 29% expected; Fisher's exact test $p < 2.2 \times 10^{-16}$), while SVAs were significantly underrepresented (17% observed vs 71% expected; Fisher's exact test $p < 2.2 \times 10^{-16}$). Importantly, 91% of the 256 human-specific TEs identified as accessible in the 5-day protocol were also found accessible in CNCCs derived using the alternative protocol. This consistency suggests that our findings are robust and independent of the CNCC differentiation method used.

## Co-opted TEs are enriched with the CNCC coordinator motif

Next, we explored the genomic features driving the recruitment of the LTR5Hs and SVAs as CNCC enhancers. First, we performed computational DNA motif analysis on the set of accessible TEs using the non-accessible LTR5Hs and SVAs as background control for differential enrichment. Recent studies have identified a specific DNA motif, known as coordinator, which is enriched at enhancers critical for the regulation of CNCC identity (Kim et al, 2024; Prescott et al, 2015). Coordinator is a composite motif which consists of a fusion between a generic AT-rich homeobox motif (TTAATTA) and the binding motif of the CNCC master regulator TWIST1, typically joined by a stretch of A nucleotides (Kim et al, 2024; Prescott et al, 2015). Importantly, we found both components of the coordinator motif as highly enriched in the set of accessible LTR5Hs and human-specific SVAs (Fig. 2D; Dataset EV3). Specifically, the TWIST1 motif and the AT-rich homeobox motif were found in 83 and 62% of accessible human-specific TEs, respectively, with similar numbers observed when incorporating non-human-specific SVAs in the analysis (Fig. 2D; Appendix Fig. 2D,E; Dataset EV3). Moreover, we found that 29% of the accessible LTR5s and SVAs harbored the full coordinator motif sequence, as opposed to only 14% of the non-accessible TEs. This suggests that the human-specific TEs accessible in CNCCs are significantly more enriched for the coordinator motif relative to the non-accessible ones (Fisher's exact test p < 0.00001).

In addition, the motifs for other CNCC signature transcription factors, such as AP2α (TFAP2A), SOX9/10, SLUG, and FOXD3,

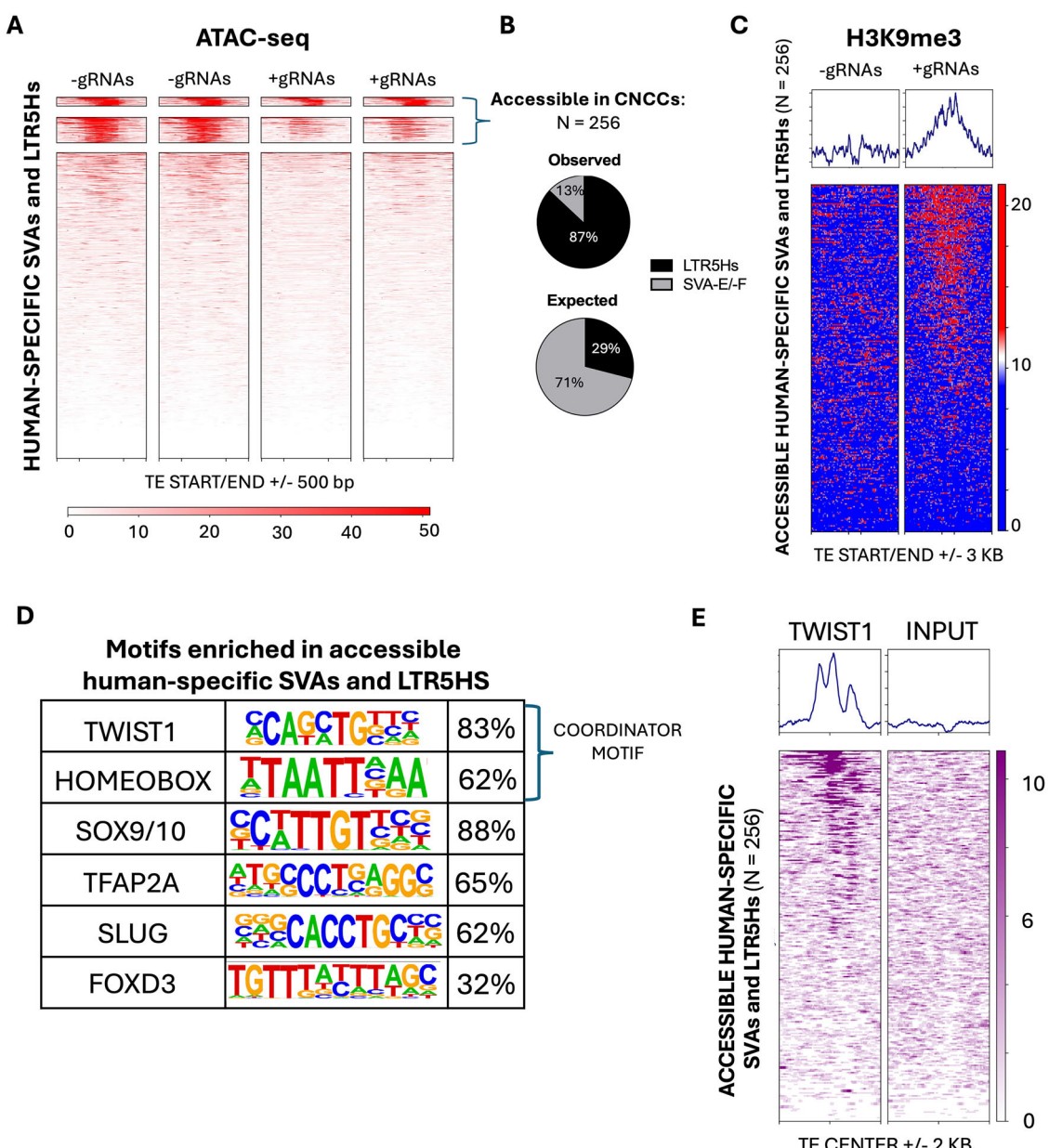

**Figure 2. Hundreds of human-specific SVAs and LTR5Hs function as enhancers in human CNCCs.**

(A) ATAC-seq signal across human-specific SVAs and LTR5Hs. Heatmap rows represent individual transposable elements (TEs) aligned at their start or end (±500 bp), and the intensity reflects chromatin accessibility. Clusters were generated with deepTools using the k-mer algorithm. Two independent replicates per group. (B) Pie charts showing the observed versus expected fractions of accessible TEs belonging to the LTR5Hs/SVA subfamilies. (C) H3K9me3 enrichment at the accessible human-specific SVAs and LTR5Hs. The top panel displays the average H3K9me3 signal (average profile) aligned to the TE start or end (±3 kb), while the heatmap below shows individual TEs ranked by signal intensity. (D) Transcription factor motif analysis of accessible human-specific SVAs and LTR5Hs. Representative enriched motifs (TWIST1, HOMEOBOX, SOX9/10, AP2α, SLUG, and FOXD3) in accessible human-specific TEs are shown with the percentage of TEs containing each motif. (E) Heatmaps of TWIST1 ChIP-seq signal (left) and corresponding input control (right) centered on the accessible human-specific SVAs and LTR5Hs (±2 kb). Color intensity indicates normalized enrichment, highlighting TWIST1 occupancy at these accessible elements. Statistical analysis on the observed and expected percentages was performed using Fisher's exact test. Source data are available online for this figure.

were also found as significantly enriched (Fig. 2D; Appendix Fig. 2D,E; Dataset EV3). Given the prevalence of the coordinator motif in accessible LTR5Hs and SVAs, we investigated whether these TEs are directly bound by TWIST1 in human CNCCs. To address this, we leveraged publicly available TWIST1 ChIP-seq data generated in iPSC-derived CNCCs (Kim et al, 2024; Data ref: Kim et al, 2024), which revealed TWIST1 ChIP-seq signal at over half of the accessible LTR5Hs and SVA in human CNCCs (Fig. 2E).

Overall, these findings suggest that DNA sequence and transcription factor binding are a major driver for TE co-option as active CNCC enhancers.

Finally, we examined whether the LTR5Hs and SVAs accessible in CNCCs integrated into previously existing enhancers or transcriptionally inactive genomic regions. To investigate this, we analyzed regions extending 250 bp upstream and downstream of these accessible TEs, assessing their accessibility in chimpanzee iPSC-derived CNCCs using publicly available paired-end ATAC-seq data (Prescott et al, 2015; Data ref: Prescott et al, 2015). Notably, we observed no accessibility signal in any of these regions (Appendix Fig. S3), suggesting that the hominoid-specific transposons likely established novel cis-regulatory elements rather than integrating into preexisting active enhancer regions.

## Repression of LTR5Hs and SVAs impairs expression of CNCC-migration genes

Our experiments so far identified ~515 LTR5Hs and SVAs (half of which are human-specific) displaying active enhancer signature in human CNCCs. Since our CRISPRi system enables the simultaneous repression of ~75% of these elements, we next investigated whether their repression had significant effects on the CNCC transcriptome. To this end, we differentiated our doxycycline-treated CRISPRi lines (+ and −gRNAs) into CNCCs and performed RNA-seq. First, we analyzed gene expression in relation to the nearest transcription start sites (TSS) of the 256 accessible human-specific SVAs and LTR5Hs. In total, we identified 107 genes located near these elements that were actively expressed in CNCCs (median TPM >1 across all −gRNA replicates). Overall, the expression of these genes was significantly decreased upon TE repression (Wilcoxon's rank-sum test $p < 0.037$; Appendix Fig. S4A), with 83/107 (= 77.5%) displaying lower expression in the +gRNA sample relative to the −gRNA counterpart (Dataset EV4). Consistent with LTR5Hs playing a prominent role in CNCC, 71 of the 83 genes (=86%) with decreased expression upon CRISPRi were the nearest gene to an accessible LTRH5s.

We leveraged publicly available RNA-seq data from CNCCs derived from chimpanzee iPSCs (Prescott et al, 2015), which we reanalysed using our pipeline, to compare the expression of these 107 genes between the two closely related ape species. This analysis revealed that these genes are typically expressed at significantly higher levels in humans than in chimpanzees (Fig. 3A; Dataset EV4). However, repressing the LTR5Hs and SVAs abolished most of this expression difference between the two species (Fig. 3A), suggesting a direct role for human-specific TEs in species-specific CNCC gene regulation. Consistent with this, repressing the hominoid-specific TEs increased the correlation between human and chimpanzee expression levels for these genes (Appendix Fig. S4B).

Next, we examined transcriptome-wide effects. Differential gene expression analysis identified 795 genes that were significantly differentially expressed between +gRNA and −gRNA CNCCs (FDR <0.05; FC<−1.5 or >1.5; Fig. 3B; Dataset EV5). Of these, 501 genes were downregulated, while 294 genes were upregulated in the +gRNA samples (Fig. 3B). Gene ontology enrichment analysis of the downregulated genes revealed a significant enrichment for cell migration-related processes, suggesting that many of these genes play key roles in CNCC migration (Fig. 3C; Appendix Fig. S5). In line with these findings, gene ontology enrichment analysis performed on the 107 expressed genes located near accessible LTR5Hs and human-specific SVAs also identified cell migration as one of the top-5 enriched pathways, despite the small sample size (Appendix Fig. S6). Conversely, upregulated genes were primarily

associated with mitochondrial and cell division processes (Appendix Fig. S7). It is worth noting that only three of the differentially expressed genes had an intronic SVA or LTR5Hs, suggesting that the high number of differentially expressed genes is not an artifact of CRISPRi-mediated repression at gene bodies.

## Silencing hominoid-specific TEs functionally affects CNCC migratory potential in vitro

Since cell migration was the predominant signature enriched in the downregulated genes, we investigated whether repressing hominoid-specific SVAs and LTR5Hs could functionally affect CNCC migration. To test this, we performed a transwell migration assay, in which hiPSC-CNCCs were seeded on geltrex-coated transwell membranes overnight and subsequently exposed to medium supplemented with the general chemoattractant fetal bovine serum (FBS) in the lower chamber (Fig. 4A). After 24 h of incubation, comparison of the −gRNA and +gRNA lines revealed a significant reduction in CNCC migration across the membrane upon repression of hominoid-specific TEs, suggesting impaired migratory capacity (Fig. 4B,C). Additionally, we performed a wound healing assay, which showed significantly impaired wound closure of the +gRNA CNCCs compared to −gRNA CNCCs when exposed to medium + 10% FBS for 8 h (Fig. 4D,E). Altogether, these findings are consistent with the RNA-seq data and suggest that SVAs and LTR5Hs-driven regulatory activity contributes to regulate human CNCC migration.

## The LTR5Hs and SVAs active in CNCCs are silent in most human cell types

Finally, we investigated whether the 256 human-specific SVAs and LTR5Hs with active enhancer signature in CNCCs were also showing cis-regulatory signature in other human cell types. To address this, we leveraged publicly available H3K27ac ChIP-seq data from 14 cell types generated by the Roadmap Epigenomics Consortium, including distinct brain regions, lung, aorta, adipose tissue, pancreas and spleen (Kundaje et al., 2015; Data ref: Kundaje et al, 2015). This analysis revealed that, in addition to CNCCs, these 256 TE exhibit active enhancer signature exclusively in endomesodermal cells, with a subset also active in iPSCs (Fig. 5). In contrast, they remain completely silenced in all other cell types (Fig. 5). The activation of these elements in endomesodermal cells is unsurprising, as a previous study from our lab has shown that human-specific SVAs are highly enriched for the motif of the key mesodermal regulator EOMES ( = TBR2; Patoori et al, 2022). Similarly, the co-option of LTR5Hs and SVAs as regulatory elements in human iPSCs and ESCs has been suggested by previous research (Barnada et al, 2022; Fuentes et al, 2018; Pontis et al, 2019).

In summary, these findings support the notion that TE co-option as functional cis-regulatory elements is largely a cell-type-specific process, with distinct subsets of human-specific TEs contributing to regulatory networks in different developmental contexts. However, a more comprehensive analysis of additional cell types should be performed to further support this model.

# Discussion

Transposable elements (TEs) have historically been viewed as genomic parasites with little relevance beyond their self-propagation

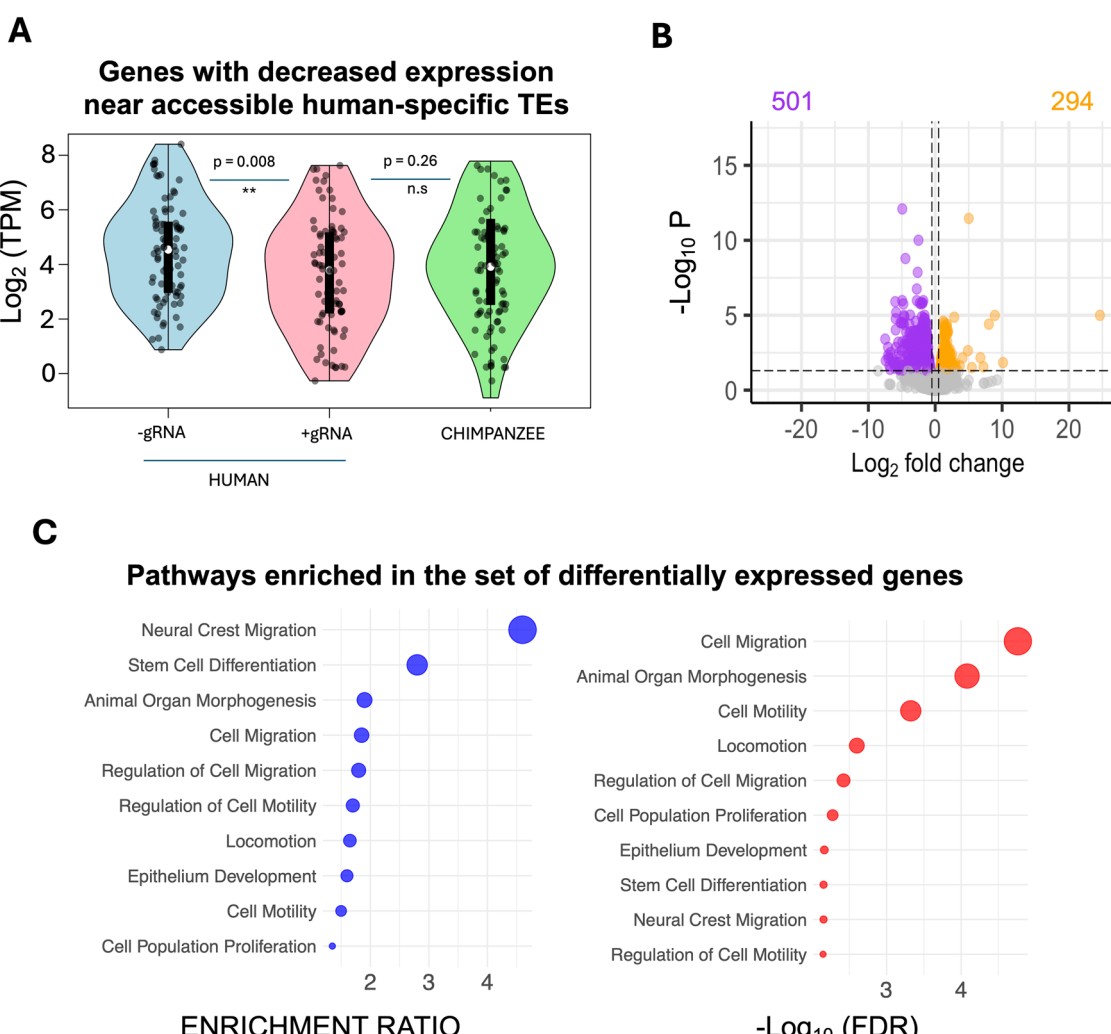

**Figure 3. Human-specific SVAs and LTR5Hs regulate the expression of CNCC migration genes.**

(A) Violin plots displaying expression (log2 TPM) of genes located near accessible human-specific SVAs and LTR5Hs. Three conditions are shown: human CNCCs generated using CRISPRi-iPSC line without guide RNAs (−gRNA), human CNCCs generated using CRISPRi-iPSC line with guide RNAs (+gRNA), and chimpanzee CNCCs. The decreased expression in the +gRNA condition indicates potential regulatory contributions of these human-specific TEs. (B) Volcano plot illustrating differentially expressed genes between the −gRNA (n = 3) and +gRNA (n = 2) conditions in human CNCCs. The x-axis shows the log2 fold change, while the y-axis represents the −log10 P value. Significantly upregulated (green) and downregulated (red) genes are highlighted, with total numbers indicated. Dashed lines mark common significance thresholds. (C) Functional enrichment analysis of downregulated genes. Bubble plots display top enriched pathways, with the x-axis indicating enrichment ratio (blue) or the −log10 FDR (red) and the y-axis listing pathway terms. The size of each bubble reflects statistical significance. Pathways involved in cell migration and motility are among the most prominently enriched. Statistical analysis was performed with the Wilcoxon rank-sum test. Source data are available online for this figure.

mechanisms. Yet, over the past two decades, numerous studies have shown that TEs can substantially influence gene regulatory architectures, driving lineage-specific developmental programs and morphological variation (Chuong et al, 2017; Feschotte, 2008; Fueyo et al, 2022; Patoori et al, 2022; Sundaram and Wysocka, 2020; Trizzino et al, 2017). In this regard, the vertebrate cranial neural crest cell (CNCC) population, which plays a pivotal role in craniofacial morphogenesis, offers an especially intriguing system for the investigation of TE co-option as a source of regulatory novelty (Bronner & LeDouarin, 2012; Gokhman et al, 2021; Minoux and Rijli, 2010; Prescott et al, 2015). Over the course of vertebrate evolution, craniofacial diversity has been shaped by small but potent modifications in the gene regulatory circuits of the neural crest (Gokhman et al, 2021; Prescott et al, 2015).

In this context, our findings suggest that hominoid-specific TEs contribute to these modifications by providing new enhancer platforms that alter CNCC gene expression, pointing to a direct role for TEs in the emergence of human-specific craniofacial features (Gokhman et al, 2021; Prescott et al, 2015).

CNCCs are a multipotent, migratory cell population that emerges from the border region of the neural tube and subsequently disperses into developing craniofacial structures, the peripheral nervous system, and other tissues (Simões-Costa et al, 2015; Theveneau and Mayor, 2012). Understanding how species-specific genomic elements, such as hominoid-specific TEs, modulate CNCC development can shed light on the evolutionary mechanisms underlying morphological divergence, particularly in

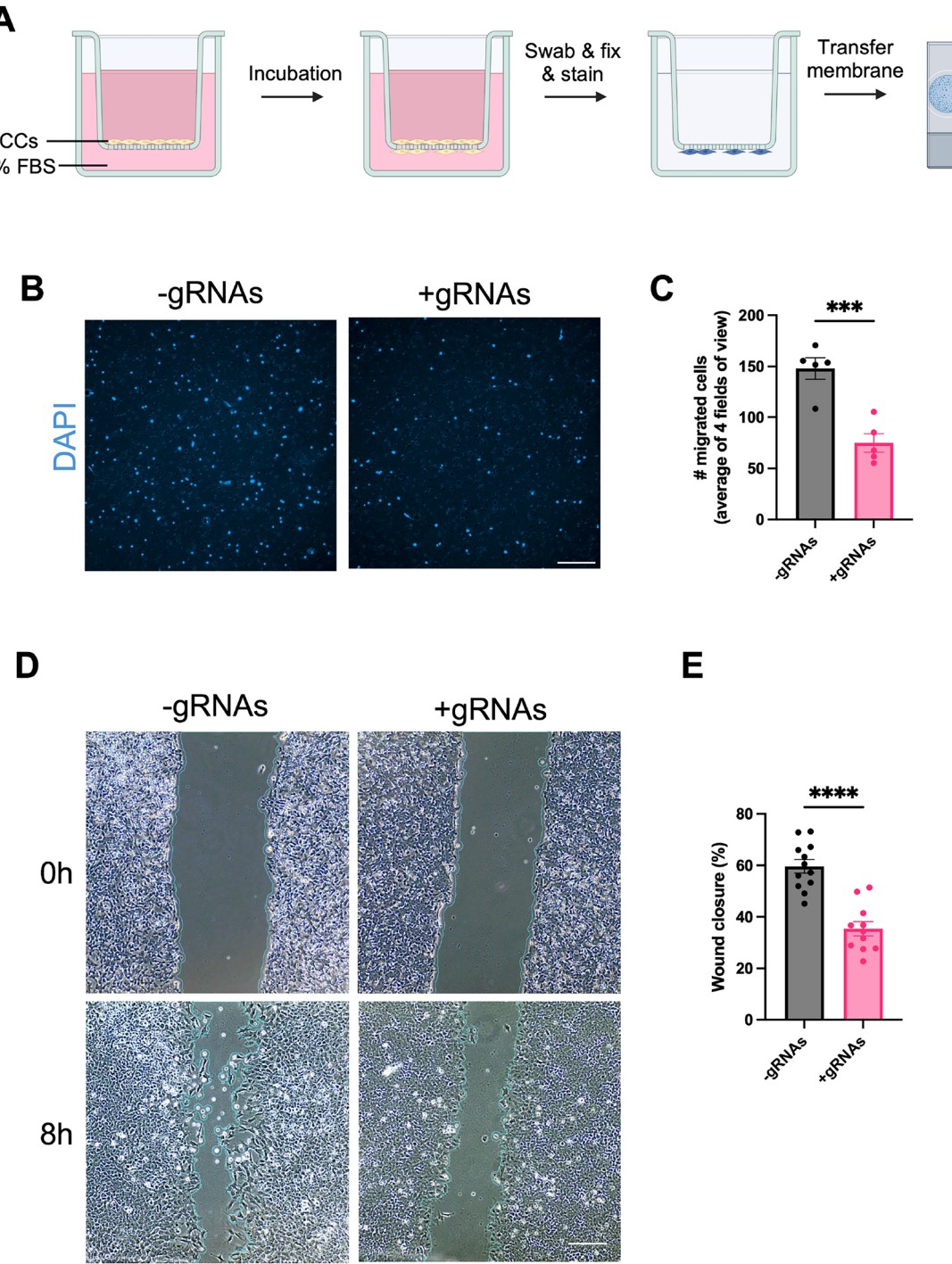

**Figure 4. HiPSC-NCCs show inferior migration capability when human-specific TEs are silenced.**

(A) Outline of the transwell migration assay. (B) $1 \times 10^5$ hiPSC-NCCs (±gRNAs) were seeded overnight, followed by a media change in the upper chamber and incubation with differentiation medium +10% FBS in the lower chamber for 24 h. At 24 h, non-migrated hiPSC-NCCs on the top of the membrane were swabbed away and migrated hiPSC-CMs were fixed, stained and imaged. DAPI staining was used to visualize nuclei. Scale bar: 50 μm. (C) Quantification of transwell membranes. 10X images were taken from four different fields of view, and the average number of migrated cells was calculated. $n = 5$. ***$p = 0.0008$. (D) $5 \times 10^4$ hiPSC-NCCs (±gRNAs) were seeded overnight onto a 24-well plate. The next day, scratch wounds were made, and the hiPSC-NCCs were incubated with differentiation medium + 10% FBS. After 8 h, scratch wounds were imaged, and wound closure was measured. $n = 12$ (−gRNAs), $n = 11$ (+gRNAs). Scale bar: 200 μm. (E) Quantification of scratch wound closure was performed with an automated plugin (Suarez-Arnedo et al, 2020). ****$p = 2.56 \times 10^{-6}$. Both assays were independently performed twice. Statistical analyses were performed using an unpaired $t$-test. Error bars indicate mean ± SEM. Source data are available online for this figure.

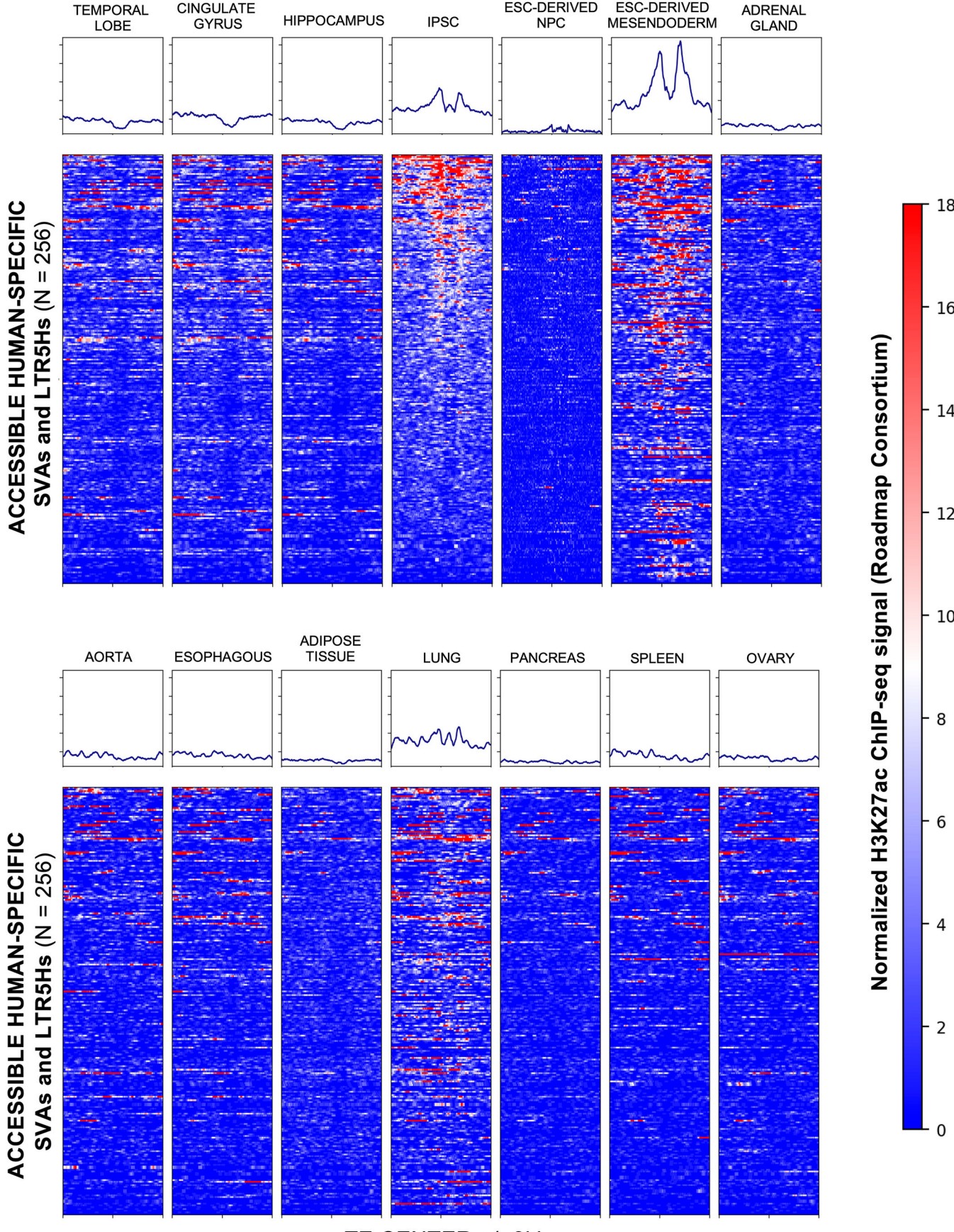

Figure 5.   Co-option of SVAs and LTR5Hs as active enhancers is cell-type specific.

Heatmaps of normalized H3K27ac ChIP-seq signal (Roadmap Epigenomics Consortium) across 256 accessible human-specific SVAs and LTR5Hs in multiple tissues and cell types. Each row corresponds to an individual transposable element (TE), aligned at its center (±3 kb), while columns represent different tissues/cell lines. The color bar on the right indicates the relative intensity of the H3K27ac signal, with higher enrichment in red and lower enrichment in blue.

the craniofacial region (Capra et al, 2013; Prescott et al, 2015). Specifically, in this study, we demonstrate that LTR5Hs and SVAs, two families of TEs unique to hominoids, function as enhancers within human CNCCs. Using integrative approaches that combined chromatin accessibility profiling, transcriptomic analysis, and functional perturbation via CRISPR-interference, we reveal that hundreds of these TEs display active enhancer signatures and exert direct regulatory control over genes crucial for CNCC migration. These findings are in line with previous studies showing that TEs can be repurposed as developmental enhancers, thereby contributing to species-specific gene regulatory landscapes in primates (Jacques et al, 2013; Patoori et al, 2022; Trizzino et al, 2017). Our data also implicate these TE-derived enhancers in driving human-specific craniofacial regulatory programs, as evidenced by the fact that repressing LTR5Hs and SVAs reduces the expression of dozens of genes in human CNCCs to levels normally observed in chimpanzees. Notably, LTR5Hs and SVAs have historically been linked to regulatory innovation in primates, though their roles in neural crest biology have remained unexplored (Fuentes et al, 2018; Imbeault et al, 2017; Patoori et al, 2022; Pontis et al, 2019).

One of the central questions in evolutionary developmental biology is how minor alterations in deeply conserved developmental pathways can yield significant morphological divergence (Carroll, 2005; King and Wilson, 1975). The CNCC population, which is indispensable for craniofacial bone and cartilage formation, serves as an excellent proof of principle in this regard. It has been suggested that relatively small changes in CNCC gene expression can profoundly impact facial shape and size (Preprint: Khouri-Farah et al, 2025; Le Douarin & Dupin, 2018; Minoux and Rijli, 2010; Mitchell et al, 2025; Prescott et al, 2015). Our study extends this framework by proposing that TEs represent a flexible reservoir of regulatory motifs capable of integrating into preexisting gene regulatory networks, fine-tuning CNCC specification and migration. These findings are consistent with previous reports showing that TEs have been exapted into tissue-specific enhancers in various lineages, including immune cells, brain cell-types, embryonic stem cells, and endometrial tissue (Chuong et al, 2017; Frost et al, 2023; Lynch et al, 2011; Trizzino et al, 2018; J. Wang et al, 2014).

Importantly, while most of the LTR5Hs elements are present as "solo LTRs" in the human genome, a few of them, also targeted by our gRNAs, encode full-length proviral sequences that express Rec, Env and Np9 subunits. Previous studies have shown that the Rec and Env subunits can effectively influence cell migration, for instance, in melanocytes in melanoma (Singh et al, 2020). Therefore, we cannot rule out that the impairment of CNCC migration is not only due to repression of the LTR5Hs-derived enhancer activity, but also to the repression of the encoded proteins of the full-length proviral sequences.

While correlative evidence can implicate TEs as enhancers, functional validation is key to establishing their regulatory impact. By employing a CRISPRi approach that targeted ~75% of all the LTR5Hs and SVAs active in CNCCs, we observed pronounced effects on gene expression with a strong enrichment for processes related to

cell migration, which is indispensable for CNCCs to navigate into the developing facial primordia (Theveneau and Mayor, 2012). The accessible LTR5Hs and SVAs that remained unaffected by CRISPRi, as they were not targeted by our gRNAs, comprised only 68 elements. Analysis of the nearest genes associated with these elements did not reveal enrichment for any specific biological pathways. Nevertheless, motif enrichment analysis against scrambled backgrounds identified significant enrichment for binding motifs of key CNCC transcription factors, including TFAP2A, SOX9/10, and TWIST1. These findings suggest that a small subset of TEs potentially bound by NCC transcription factors may escape effective silencing in our experimental system, although they represent a minority of elements.

Overall, our results not only reinforce the notion that TEs influence CNCC gene networks but also demonstrate that their regulatory input is significant enough to modulate key developmental pathways. This model agrees with evidence from other contexts in which TEs have driven evolutionary innovations. For instance, MER41 elements in the human genome have been exapted as enhancers regulating immune responses (Chuong et al, 2017). Similarly, in the endometrium and placenta, endogenous retroviral LTRs have been implicated in the evolution of pregnancy-specific gene regulatory networks (Frost et al, 2023; Lynch et al, 2011). Our study adds craniofacial development to the growing list of developmental systems shaped by TE co-option, further highlighting the broad evolutionary relevance of these mobile elements.

TE co-option relies on the inherent or mutated presence of transcription factor binding sites within TE sequences (Barnada et al, 2022; Bejerano et al, 2006; Feschotte, 2008; Patoori et al, 2022; Trizzino et al, 2017, 2018). In this study, we show that LTR5Hs and SVAs domesticated as enhancers in CNCCs harbor binding sites for the CNCC coordinator motif (Kim et al, 2024; Prescott et al, 2015), as well as motifs for TFAP2A, SOX9/10 and other key regulators of neural crest identity. We hypothesize that these factors bind cooperatively to multi-part motifs within the TE loci, enabling the formation of enhancer complexes that drive spatial and temporal gene expression programs in CNCCs. Notably, TFAP2A expression was reduced, albeit modestly, upon TE repression. TFAP2A is critical for CNCC specification and migration, and it is possible that dysregulation of this factor could, at least in part, underpin some of the aberrant migratory phenotypes.

Disruptions to CNCC gene regulatory networks can underlie congenital anomalies such as cleft lip and palate, craniosynostosis, and other craniofacial malformations (Preprint: Khouri-Farah et al, 2025; Trainor, 2010). Given that TE-derived enhancers are integral to the CNCC transcriptional program, mutations or epigenetic dysregulations at these loci could contribute to such disorders. Future studies investigating the links between TE variations and craniofacial pathologies may reveal novel diagnostic markers or therapeutic targets.

In addition, comparative investigations in other hominoids, including gorillas and orangutans, could help unravel whether other TEs have been similarly co-opted in the cranial neural crest. Such research would clarify whether TE-driven craniofacial

enhancer innovations are a hallmark across all hominoids or largely unique to the human lineage, shedding light on how small-scale genetic elements can produce macro-evolutionary changes in form and function.

Finally, our study has some limitations. Firstly, employing distinct gRNAs for LTR5Hs and SVAs would be ideal to more precisely differentiate the roles of these two TE families in CNCC development. Additionally, we assigned genes to TE-derived enhancers based on genomic proximity, but enhancer-promoter contact maps would offer more precise enhancer-gene associations. Unfortunately, genome-wide contact maps are currently unavailable for CNCCs. Relatedly, the expression of a single gene is often regulated by multiple redundant enhancers, complicating the precise determination of individual enhancer contributions without CNCC-specific enhancer-promoter contact data. Consequently, the absence of a detectable gene expression change upon CRISPRi does not necessarily exclude enhancer activity of a specific TE, as compensatory enhancers might mask the loss-of-function phenotype.

Furthermore, while our Gene Ontology analysis robustly highlighted a "cell migration" signature, functionally validated through transwell and wound healing assays, we could not definitively identify the exact genes responsible for the altered migratory phenotype observed upon repression of hominoid-specific TEs.

Another limitation is our reliance on uniquely mapping sequencing reads. Although this approach ensures specificity, it likely omits biologically meaningful signals from multi-mapping reads. Lastly, while we hypothesize that the observed phenotype arises from the co-option of TEs into CNCC migration-associated enhancers, additional validation via techniques like ChIP-STARR-seq combined with targeted genetic perturbations would help substantiate this model and confirm that disruption of these enhancers directly influences gene expression and related cellular behaviors.

In conclusion, our findings provide further evidence for TEs as highly adaptive genomic elements that actively contribute to shaping developmental and evolutionary trajectories. Through combined roles in CNCC specification, migration, and potentially other lineage-specific developmental programs, TEs demonstrate how evolutionary tinkering at the regulatory level can yield morphological diversity.

# Methods

### Reagents and tools table

| Reagent/resource | Reference or source | Identifier or catalog number |
|---|---|---|
| **Experimental models** | | |
| CRISPRi hiPSC line (−gRNAs) | This study. Manufactured by Applied StemCell | C2277_B8 |
| CRISPRi hiPSC line (+gRNAs) | This study. Manufactured by Applied StemCell | C2277_A1 |
| **Antibodies** | | |
| Mouse AP2α (3B5) | Fisher Scientific | 11594723 |
| Rabbit Sox9 (EPR14335) | Abcam | AB185230-1001 |

| Reagent/resource | Reference or source | Identifier or catalog number |
|---|---|---|
| Goat anti-Mouse IgG2b Cross-Adsorbed Secondary Antibody, Alexa Fluor™ 647 | Life Technologies | A21242 |
| Goat anti-Mouse IgG2b Cross-Adsorbed Secondary Antibody, Alexa Fluor™ 647 | Life Technologies | A21242 |
| Mouse Cas9 | Cambridge Bioscience | 61757 |
| Rabbit GAPDH (D16H11) | Cell Signalling Technology | 5174 |
| Goat anti-mouse IgG IRDye® 800CW | LI-COR | 926-32210 |
| Goat anti-Rabbit IgG IRDye® 800CW | LI-COR | 926-32211 |
| Rabbit H3 tri methyl K9 - ChIP Grade | Abcam | ab8898 |
| **Oligonucleotides and other sequence-based reagents** | | |
| Primer name | Sequence (5'-3') | |
| SOX9 | F: 5'-GTACCCGCACTTGCACAAC-3' R: 5'-TCTCGCTCTCGTTCAGAAGTC-3' | |
| SOX10 | F: 5'-GAGGGCTCCCCCATGTCAGAT-3' R: 5'-GTCTGCCTTGCCCGACTGC-3' | |
| TFAP2A | F: 5'-GCCTCTCGCTCCTCAGCTCC-3' R: 5'-CGTTGGCAGCTTTACGTCTCCC-3' | |
| TWIST1 | F: 5'-GCCAGGTACATCGACTTCCTCT-3' R: 5'-TCCATCCTCCAGACCGAGAAGG-3' | |
| GAPDH | F: 5'-GAACGGGAAGCTTGTCATCAA-3' R: 5'-ATCGCCCCACTTGATTTTGG-3' | |
| **Chemicals, enzymes and other reagents** | | |
| mTeSR Plus | StemCell Technologies | 100-0276 |
| Geltrex | Thermo Fisher | A1413302 |
| DMEM/F12 | Gibco | A4192002 |
| B-27 Supplement | Fisher Scientific | 15717988 |
| CHIR99021 | Stratech | S1263-SEL-5mg |
| Bovine Serum Albumin | Fisher Scientific | 12881630 |
| GlutaMAX Supplement | Gibco | 35050061 |
| Penicillin-Streptomycin | Gibco | 15070063 |
| Y-27632 | Cambridge Bioscience | HY-119937-5MG |
| Doxycycline | Bio-Techne | 4090/50 |
| StemPro Accutase | Gibco | A1110501 |
| Triton X-100 | Merck | 648463 |
| Donkey Serum | Abcam | AB7475 |
| DAPI | BioLegend | 422801 |
| Tween:PBS | Promega | H5152 |
| RIPA Buffer | Fisher Scientific | 10230544 |
| Pierce Protease Inhibitor Tablet | Fisher Scientific | 15614189 |
| Pierce BCA Protein Assay Kit | Fisher Scientific | 23225 |

| Reagent/resource | Reference or source | Identifier or catalog number |
|---|---|---|
| Laemmli Buffer | Fisher Scientific | J61337.AC |
| Novex Tris-Glycine Mini Protein Gel (4–12%) | Invitrogen | XP04122BOX |
| Novex Tris-Glycine SDS Running Buffer | Fisher Scientific | 11559066 |
| NuPAGE Transfer Buffer | Life Technologies | NP00061 |
| Intercept Blocking Buffer | Li-Cor | 927-70001 |
| Monarch Total RNA Miniprep Kit | New England BioLabs | T2010 |
| Maxima First Strand cDNA Synthesis Kit | Thermo Fisher | K1641 |
| PowerUp SYBR Green Supermix | Life Technologies | A46112 |
| 24-well inserts (8 µm) | Starstedt | 83.3932.800 |
| NEBNext Poly(A) mRNA Magnetic Isolation Module | New England BioLabs | E7490 |
| NEBNext Ultra II Directional RNA Library Prep Kit | New England BioLabs | E7760 |
| Dynabeads Protein G magnetic beads | Invitrogen | 10003D |
| H3K9me3 Antibody | Abcam | ab8898 |
| Zymo ChIP DNA Clean and Concentrator Kit | Zymo Research | D5205 |
| Quantifluor ONE dsDNA system | Promega | E4871 |
| NEBNext Ultra II DNA Library Prep Kit | New England BioLabs | E7645L |
| ATAC-Seq Kit | Active Motif | 53150 |
| SPRI Beads | Beckman Coulter | B23318 |
| Dual Index Primers | New England BioLabs | E7600S |
| **Software** | | |
| Kallisto 0.46.2 | Bray et al, 2016 | |
| DESeq2 1.28.0 | Love et al, 2014 | |
| Webgestalt 2019 | https://webgestalt.org Liao et al, 2019 | |
| BWA alignment | Li & Durbin, 2009 | |
| HOMER | Heinz et al, 2010 | |
| BEDTools 2.31.1 | Quinlan and Hall, 2010 | |
| deepTools 3.5.5 | Ramírez et al, 2014 | |
| SAMTools 1.18 | Li et al, 2009 | |
| TE transcripts 2.2.3 | Jin et al, 2015 | |
| GraphPad Prism 10 | https://www.graphpad.com | |
| FIJI (ImageJ) | https://fiji.sc | |
| **Other** | | |
| Illumina NovaSeq X Plus | Illumina | |
| Tapestation 2200 | Agilent | |
| Covaris M220 Focused-Ultrasonicator | Covaris | |

## Generation of doxycycline-inducible dCas9-KRAB hiPSC lines

A plasmid containing a tetracycline-inducible dCas9-KRAB expression cassette flanked by piggyBac recombination sites was obtained from the Wysocka Lab at Stanford University. This plasmid, referred to as "p-dCas9-KRAB," provides constitutive puromycin resistance, enabling selection of stable clones when co-expressed with the piggyBac transposase plasmid ("p-PB-Transposase," Systems Bioscience). For the gRNAs line, the C2277 hiPSC line was co-transfected with p-dCas9-KRAB and p-PB-Transposase by Applied StemCell Inc. (C2277-A, −gRNAs line). Next, a piggyBac transposon plasmid ("p-sgRNA," Systems Bioscience) containing two sgRNAs targeting ~80% of annotated SVAs and LTR5Hs in humans was obtained based on a previously published study (Pontis et al, 2019). This plasmid confers constitutive dual sgRNA expression and geneticin resistance. For the +gRNAs line, the C2277 hiPSC line was co-transfected with p-dCas9-KRAB, p-PB-Transposase, and p-sgRNA (C2277-B, +gRNA line). Puromycin (0.125 µg/mL) only (−gRNAs line) or puromycin (0.125 µg/mL) and geneticin (100 µg/mL) (+gRNAs line) were introduced for multiple days to obtain purified colonies. Both cell lines were authenticated by STR profiling and screened for mycoplasma contamination prior to receipt.

## iPSC culture and NCC differentiation

Human iPSC lines C2277-A (−gRNAs) and C2277-B (+gRNAs) were cultured in feeder-free, serum-free 2D culture at 37 °C and 5% $CO_2$. For the generation of NCCs, the differentiation method was derived from the protocol published by Leung et al (2016). Briefly, hiPSCs were grown in mTeSR Plus (StemCell Technologies, 100-0276) on Geltrex-coated (Thermo Fisher, A1413302) six-well plates until reaching ~80% confluency. Differentiation was initiated by plating $2 \times 10^6$ hiPSCs in Neural Crest Differentiation media (DMEM/F12 (Gibco, A4192002), 2% B-27 supplement (Fisher Scientific, 15717988), 3 µM CHIR99021 (Stratech, S1263-SEL-5mg), 0.5% bovine serum albumin (Fisher Scientific, 12881630), 1X glutaMAX supplement (Gibco, 35050061), 1% penicillin-streptomycin (Gibco, 15070063)) onto Geltrex-coated six-well plates supplemented with 10 µM Y-27632 (Cambridge Bioscience, HY-119937-5MG). The following days, daily media changes were performed, and 2 µg/mL doxycycline was added to the culture media from day 2 onwards (i.e., 24 h after the start of differentiation). NCC differentiation was assessed after 5 days.

## Immunocytochemistry

HiPSCs-NCCs were harvested by incubation with StemPro Accutase (Gibco, A1110501) for 5–10 min, followed by aspiration and detachment with DMEM/F12, and subsequently seeded onto 16 mm coverslips in 12-well plates coated with Geltrex and left overnight to attach before fixation with 4% paraformaldehyde/PBS for 15 min at 37 °C. After washing with PBS, the fixed cells were stored in PBS until staining. HiPSC-NCCs were subsequently permeabilised with 0.1% Triton X-100/PBS (Merck, 648463) for

10 min at RT and blocked using 10% donkey serum/PBS (Abcam, AB7475) for 1 h at RT. Cells were stained with primary antibodies overnight at 4 °C in the dark, washed and stained with secondary antibodies for 30 min at 37 °C. The antibodies used can be found in the Reagents and Tools Table. After a subsequent washing step, cells were counterstained with 1 μg/mL DAPI (BioLegend, 422801). Washing steps were performed with 0.1% Tween:PBS (Promega, H5152). Images were acquired using a Zeiss Axio Observer, and further processing was done using ImageJ. Normalized intensity was calculated by dividing the signal intensity of each frame by the number of cells.

## Western blot

Cells were lysed in radioimmunoprecipitation assay (RIPA) buffer (Fisher Scientific, 10230544) supplemented with a Pierce protease inhibitor tablet (Fisher Scientific, 15614189). The lysates were vortexed for 30 s and incubated on ice for 30 min. Samples were then centrifuged at $17,000 \times g$ for 10 min at 4 °C, and the supernatant was collected. Protein concentration was determined using a Pierce BCA Protein Assay Kit (Fisher Scientific, 23225) according to the manufacturer's protocol. The plate was incubated at 37 °C for 30 min and analyzed using the SpectraMax M2 microplate reader (Molecular Devices) at 562 nm. About 10 ug of protein were denatured in Laemmli Buffer (Fisher Scientific, J61337.AC) at 95 °C for 5 min. Proteins were separated using a 4–12% Novex Tris-Glycine Mini Protein Gel (Invitrogen, XP04122BOX) in Novex Tris-Glycine SDS Running Buffer (Fisher Scientific, 11559066) for 90 min. Proteins were transferred onto a nitrocellulose membrane using the Mini Trans-Blot Cell system with NuPAGE Transfer Buffer (Life Technologies, NP00061) for 1 h. Membranes were blocked in Li-Cor Intercept Blocking Buffer (Li-Cor, 927-70001) for 1 h at RT, then incubated overnight at 4 °C with primary antibodies (see Reagents and Tools Table) diluted in Intercept Blocking Buffer with 0.2% Tween. The following day, washes were performed in 0.1% Tween/PBS, and membranes were incubated with secondary antibodies (see Reagents and Tools Table) for 30 min before imaging on a Li-Cor Odyssey XF imaging system.

## Real-time qPCR

Total RNA was isolated using the Monarch Total RNA miniprep kit (New England BioLabs, T2010) following the manufacturer's instructions. Briefly, harvested cells were lysed in RLT buffer. The lysate was then passed through a gDNA column and centrifuged at $16,000 \times g$ for 1 min. Next, 70% ethanol was added to the lysate, which was transferred to an RNeasy silica column and centrifuged at $16,000 \times g$ for 1 min. The column was washed, followed by DNase I treatment for 15 min at RT. Additional wash steps were performed, followed by a final dry centrifugation at 12,000 rpm for 1 min. RNA was eluted in RNase-free $ddH_2O$, and centrifuged at $16,000 \times g$ for 1 min. The eluted RNA was either used immediately or stored at $-80$ °C. Reverse transcription into cDNA was performed using the Maxima First Strand cDNA Synthesis kit with the following thermal cycler settings: 10 min at 25 °C, 15 min at 50 °C, and 5 min at 85 °C. Gene expression was quantified via real-time PCR PowerUp SYBR Green Supermix (Life Technologies, A46112). Primers for each gene are listed in the Reagents and Tools

Table. GAPDH was used as a housekeeping gene. RT-qPCR was performed using a CFX Connect Real-Time PCR Detection System (Bio-Rad) with the following cycling parameters: 5 min at 95 °C, 40 cycles of 15 s at 95 °C, 30 s at 60 °C, and 30 s at 72 °C.

## Transwell migration assay

Twenty-four-well inserts (8 μM, Starstedt) were coated on both sides with ~22 μg/cm² Geltrex for 1 h at 37 °C. After removing excess liquid, $1 \times 10^5$ hiPSC-NCCs were seeded in Neural Crest Differentiation media + 10 μM Y-27632 in the upper compartment and incubated overnight for attachment. The following day, the media was replaced, and Neural Crest Differentiation media supplemented with 10% FBS was added to the lower compartment. Migration was assessed by fixing the membranes in 4% paraformaldehyde/PBS for 15 min at RT, followed by PBS washing and DAPI staining for 15 min at RT. To ensure visualization of migrated cells only, the upper surface of the insert was carefully swabbed with a cotton swab before fixation. Membranes were then excised and mounted on microscope slides for imaging. 10X images were captured from four different fields of view, and the average number of migrated cells was counted. Images were acquired using a Zeiss Axio Observer, and migration was quantified using ImageJ.

## Wound healing assay

HiPSC-NCCs were seeded into 24-well plates at a density of $2.5 \times 10^5$ cells/cm² overnight to generate confluent monolayers. A scratch wound was made with a P200 tip and following a wash with PBS, hiPSC-NCCs were incubated with Neural Crest Differentiation media + 10% FBS. After 8 h, migration was assessed, and the wounds were imaged using the EVOS XL Core Cell Imaging System. Wound closure was measured automatically using an ImageJ plugin (Suarez-Arnedo et al, 2020).

## Bulk RNA-seq library preparation

Total RNA was isolated using the Monarch Total RNA miniprep kit (New England BioLabs, T2010) following the manufacturer's instructions. RNA was quantified using a Nanodrop spectrophotometer, and RNA integrity was checked on a Tapestation 2200 (Agilent). Only samples with a RIN >7.0 were used for transcriptome analysis. RNA libraries were prepared using 300 ng of total RNA input using the NEBNext Poly(A) mRNA Magnetic Isolation Module(E7490) and NEBNext® Ultra II Directional RNA Library Prep Kit for Illumina (E7760) according to the manufacturer's instructions. PE150 sequencing was performed on a NovaSeq X Plus instrument (Illumina).

## RNA-seq analysis

Adapters were removed using TrimGalore!, and gene-level read counts were obtained with Kallisto (Bray et al, 2016). Differential gene expression analysis was performed using DESeq2 (Love et al, 2014). GO enrichment analysis was performed using Webgestalt 2019 (Liao et al, 2019). TE expression analysis was performed using TE transcripts (Jin et al, 2015). Additional statistical analysis was carried out using R (version 4.2.2) and GraphPad Prism 10.

## ChIP-seq library preparation

10 million hiPSC-NCCs were crosslinked in 1% paraformaldehyde for 5 min at RT with mild rotation. Crosslinking was quenched with 0.125 M glycine by rotation for 5 min at RT. Cells were washed twice with cold PBS and were snap-frozen for 15 min on dry ice. Samples were stored at $-80\,°C$ until further processing. Cells were resuspended in 1 mL ChIP buffer (150 mM NaCl, 1% Triton X-100, 500 mM DTT, 10 mM Tris-HCl, 5 mM EDTA) supplemented with Pierce protease inhibitor tablet and incubated on ice for 10 min. SDS was added to each sample to a final concentration of 0.3%, and chromatin was sheared using the following settings: 9 min, duty factor 5%, $6\,°C$ in a Covaris M220 Focused-Ultrasonicator. Chromatin fragment size was assessed on a Tapestation 2200.

The chromatin lysates were then centrifuged at $13,000 \times g$ for 10 min at $4\,°C$, and the supernatant was subsequently incubated with ChIP buffer supplemented with Pierce protease inhibitor and Dynabeads Protein G magnetic beads (Invitrogen) along with 3 µg of H3K9me3 antibody (Abcam, ab8898) and incubated overnight at $4\,°C$ under mild rotation. The following day, samples were placed in a magnetic rack and washed twice with Mixed Micelle Wash Buffer (150 mM NaCl, 1% Triton X-100, 0.2% SDS, 20 mM Tris-HCl, 5 mM EDTA, 65% sucrose), twice with Buffer 200 (200 mM NaCl, 1% Triton X-100, 0.1% sodium deoxycholate, 25 mM HEPES, 10 mM Tris-HCl, 1 mM EDTA), twice with LiCl/detergent wash (250 mM LiCl, 0.5% sodium deoxycholate, 0.5% NP-40, 10 mM Tris-HCl, 1 mM EDTA), once with cold TE. The beads were resuspended in TE + 1% SDS and incubated at $65\,°C$ for 10 min at 1200 rpm to elute immunocomplexes. The elution was repeated twice, and the samples were incubated overnight at $65\,°C$ to reverse crosslinking. The following day, Proteinase K (0.5 mg/mL) was added to digest samples at $65\,°C$ for 1 h. DNA was purified using the Zymo ChIP DNA Clean and Concentrator kit (Zymo, D5205) and quantified with the Quantifluor ONE dsDNA system (Promega, E4871). DNA libraries were prepared using the NEBNext Ultra II DNA library Prep Kit (E7645L) for Illumina. PE150 sequencing was performed on a NovaSeq X Plus instrument (Illumina).

## ChIP-seq analysis

Adapters were removed with TrimGalore! and reads were aligned to hg38 using the Burrows-Wheeler Alignment tool, with the MEM algorithm (Li and Durbin, 2009). Uniquely mapping reads were filtered (MAPQ >10), PCR duplicates were removed, and mitochondrial reads were discarded. Peaks were called using HOMER with default parameters at 5% FDR (Heinz et al, 2010). All statistical analyses were performed using BEDTools (Quinlan and Hall, 2010), deepTools (Ramírez et al, 2014), R (version 4.2.2) and GraphPad Prism 10.

## ATAC-seq library preparation

DNA libraries were prepared following the ATAC-Seq kit (Active Motif, 53150) according to the manufacturer's instructions. About 100,000 cells were tagmented per sample, DNA was purified using SPRI beads and amplified with dual index primers. Libraries were assessed for size distribution using the TapeStation 2200, and PE150 sequencing was performed on a NovaSeq X Plus instrument (Illumina).

## ATAC-seq analysis

Adapters were removed with TrimGalore! and reads were aligned to hg38 using the Burrows-Wheeler Alignment tool, with the MEM algorithm (Li and Durbin, 2009). With SAMTools (Li et al, 2009), uniquely mapping reads were filtered (MAPQ >10), PCR duplicates were removed, and mitochondrial reads were discarded. Consensus peaks were determined in each cell line using BEDTools (Quinlan and Hall, 2010). Motif analysis was performed using HOMER (Heinz et al, 2010). All further downstream analysis was performed using BEDTools and deepTools (Ramírez et al, 2014).

## Statistics

RT-qPCR, transwell migration and image quantification data were analysed using Graphpad Prism 10 software (Graphpad, San Diego, CA). Data were represented as mean ± standard error of mean (SEM). Statistical significance was accepted at $p < 0.05$. Sample sizes were not determined using statistical methods but were based on previous experiments under similar conditions. Blinding was not performed on experiments, but data analysis was automated for all experiments (e.g., intensity thresholds or automated plugins in FIJI). Sequencing data was analysed using BEDTools, DeepTools, and R as indicated. Motif analysis was performed using HOMER. GO enrichment analysis was performed using Webgestalt 2019.

# Data availability

RNA-seq, ATAC-seq, and ChIP-seq data have been deposited in the Gene Expression Omnibus (GEO) under accession code GEO: GSE292478. (https://www.ncbi.nlm.nih.gov/geo/query/acc.cgi?acc=GSE292478) and are publicly available as of the date of manuscript submission.

The source data of this paper are collected in the following database record: biostudies:S-SCDT-10_1038-S44320-025-00151-z.

# Peer review information

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

## Acknowledgements

The authors thank the Wysocka group at Stanford (and particularly Dr. Raquel Fueyo) for providing the dCAS9-KRAB vector, and Andrew Isopi and Dr. Samantha Barnada (Thomas Jefferson University), for designing the plasmid with the sgRNAs. We thank the Facility for Imaging by Light Microscopy (FILM) at Imperial College London for providing access to instrumentation and technical support, which is part-supported by funding from the Wellcome Trust (grant 104931/Z/14/Z) and BBSRC (grant BB/L015129/1). For this work, MT was funded by the Biotechnology and Biological Sciences Research Council (BBSRC, grant BB/Y000854/1).

## Author contributions

**Laura Deelen**: Conceptualization; Formal analysis; Investigation; Methodology; Writing—original draft. **Zoe H Mitchell**: Investigation; Writing—review and editing. **Martina Demurtas**: Investigation; Writing—review and editing. **Andria Koulle**: Investigation. **Beatriz Garcia Del Valle**: Investigation; Writing—review and editing. **Marco Trizzino**: Conceptualization; Data curation; Formal analysis; Supervision; Funding acquisition; Investigation; Methodology; Writing—original draft; Project administration.

Source data underlying figure panels in this paper may have individual authorship assigned. Where available, figure panel/source data authorship is listed in the following database record: biostudies:S-SCDT-10_1038-S44320-025-00151-z.

## Disclosure and competing interests statement

The authors declare no competing interests.

