## [Peer Review File · Molecular Systems Biology]

Hominoid-specific transposable elements reshaped neural crest migration in craniofacial development

Laura Deelen, Zoe Mitchell, Martina Demurtas, Andria Koulle, Beatriz Garcia Del Valle, and MARCO TRIZZINO

Corresponding author(s): MARCO TRIZZINO (m.trizzino@imperial.ac.uk)

Review Timeline:

Submission Date:	10th Apr 25
Editorial Decision:	17th May 25
Revision Received:	9th Jul 25
Editorial Decision:	20th Aug 25
Revision Received:	28th Aug 25
Accepted:	31st Aug 25

Editor: Yehu Moran

Transaction Report:

17th May 2025

Manuscript Number: MSB-2025-13036

Title: Human-specific transposons rewired the gene regulatory networks essential for neural crest migration

Author: Laura Deelen

Zoe Mitchell

Martina Demurtas

Beatriz Garcia Del Valle

MARCO Trizzino

Dear Dr. Trizzino,

Thank you again for submitting your work to Molecular Systems Biology. We have now heard back from the three referees who agreed to evaluate your manuscript. As you will see from the reports below, the referees find the topic of your study of potential interest. They raise, however, substantial concerns on your work, which preclude its publication in its present form and call for a major revision.

When you resubmit your manuscript, please download our CHECKLIST (<https://bit.ly/EMBOPressAuthorChecklist>) and include the completed form in your submission.

Please note that the Author Checklist will be published alongside the paper as part of the transparent process (<https://www.embopress.org/page/journal/17444292/authorguide#transparentprocess>).

If you feel you can satisfactorily address the points listed by the referees, you are welcome to submit a revised version of your manuscript. Please attach a covering letter giving a detailed account in a point-by-point format of the way in which you have handled each of the comments raised by the referees. A revised manuscript will be once again subject to external review and you probably understand that we can give you no guarantee at this stage that the eventual outcome will be favorable.

Yours sincerely,

Yehu Moran

Academic Editor

Molecular Systems Biology

We realize that it is difficult to revise to a specific deadline. In the interest of protecting the conceptual advance provided by the work, we recommend a revision within 3 months (15th Aug 2025). Please discuss the revision progress ahead of this time with the editor if you require more time to complete the revisions. Use the link below to submit your revision:

IMPORTANT: When you send your revision, we will require the following items:

1. the manuscript text in LaTeX, RTF or MS Word format
2. a letter with a detailed description of the changes made in response to the referees. Please specify clearly the exact places in the text (pages and paragraphs) where each change has been made in response to each specific comment given
3. three to four 'bullet points' highlighting the main findings of your study
4. a short 'blurb' text summarizing in two sentences the study (max. 250 characters)
5. a 'thumbnail image' (550px width and max 400px height, Illustrator, PowerPoint or jpeg format), which can be used as 'visual title' for the synopsis section of your paper.
6. Please include an author contributions statement after the Acknowledgements section (see <https://www.embopress.org/page/journal/17444292/authorguide>)
7. Please complete the CHECKLIST available at (<https://bit.ly/EMBOPressAuthorChecklist>).

Please note that the Author Checklist will be published alongside the paper as part of the transparent process (<https://www.embopress.org/page/journal/17444292/authorguide#transparentprocess>).

See also figure legend guidelines: <https://www.embopress.org/page/journal/17444292/authorguide#figureformat>

9. Please note that corresponding authors are required to supply an ORCID ID for their name upon submission of a revised manuscript (EMBO Press signed a joint statement to encourage ORCID adoption).

(<https://www.embopress.org/page/journal/17444292/authorguide#editorialprocess>)
Currently, our records indicate that the ORCID for your account is 0000-0002-1383-7200.

Please click the link below to modify this ORCID:
Link Not Available

11. Include a Reagents and Tools Table as part of the Methods section, which can be downloaded from our author guidelines (<https://www.embopress.org/page/journal/17444292/authorguide#structuredmethods>)

*** PLEASE NOTE *** As part of the EMBO Press transparent editorial process initiative (see our Editorial at <https://dx.doi.org/10.1038/msb.2010.72>), Molecular Systems Biology publishes online a Review Process File with each accepted manuscripts. This file will be published in conjunction with your paper and will include the anonymous referee reports, your point-by-point response and all pertinent correspondence relating to the manuscript. If you do NOT want this File to be published, please inform the editorial office at contact@molsystbiol.org within 14 days upon receipt of the present letter.

Reviewer #1:

This study from Marco Trizzino and colleagues finds an important and innovative evolutionary detail of cranial neural crest cell (CNCC) specification, which appears, in part, to be mediated by human-specific Endogenous Retroviruses (LTR5_Hs). These HERVs are the youngest ones in our genome that are remnants of ancient viruses that infiltrated the genomes of our ancestors after the human-chimp split (< 6 million years ago). Initially, these viral invaders might have behaved like deleterious insertions as evident from their major existence as solo-LTRs in our genome, which over time have become the special sites for regulating the expression of neighbor genes. Although a single LTR5_Hs might have a small effect, the cumulative influence of hundreds of them can dramatically impact how our genes respond during the cell specification, migration or even differentiation. The novel findings of this research are the discovery that these elements are not static relics; they have been continuously refined by evolution to select them as contributors to human-specific features. In particular, this research shows that LTR5_Hs have evolved precise TWIST1 binding motifs within the chromatin accessible regions of CNCC and demonstrate that LTR5Hs and SVAs probably function as enhancers within human CNCCs. Overlaying chromatin accessibility, transcriptomics and genetic perturbation using state-of-the-art CRISPR-interference, the authors suggest that around 500 of their loci are functional enhancers for 107 genes that plausibly lead to CNCC migration. A few of these genes are also among those whose expression is gained to be human-specific in CNCCs.

This research also answers, to some extent, why the youngest LTRs and SVAs underwent rapid expansion in the common ancestor post-split of humans and chimpanzees. This rapid expansion suggests that these elements provided ready-made genetic tools during periods of rapid environmental change, allowing our ancestors to fine-tune their cranial morphology swiftly and effectively. This adaptive mechanism, forged over millions of years, not only explains the rapid evolution of our craniofacial structures and peripheral nervous system but may also account for the genetic basis of susceptibility to related diseases in modern humans.

I like this study as it gets close to answering why there is a remarkable regulatory potential of the youngest endogenous retroelements in the human genome. It suggests that these elements may have undergone positive selection driven by their potent enhancer activity to push the neural crest cells that renovate cranial structures in conjunction with peripheral neuronal networks, thereby contributing significantly to the host's regulatory responses upon environmental challenges. This work indeed represents a significant advance over previously published studies. Thus, this line of advancement has enormous potential to outreach plenty of molecular system biology audience, genome researchers, chromatin and transposon audiences. Overall, although preliminary, the results are interesting and worthy of publishing. At this point, however, a few issues arise that need further clarification and maybe analysis before I consider this study complete.

Major Concerns:

My major concern is that the manuscript does not adequately address its limitations.

1. For example, the authors have not used activity-by-contact (ABC) maps to infer looping interactions between LTR5_HS-derived enhancers and their target gene promoters.
2. Moreover, it is unclear which specific genes are implicated and how these genes contribute to the migration of CNCCs. Authors have performed a pathway analysis on the differentially expressed genes, which is a substandard method to decipher the real physiological relevance. The investigations presented here are exploratory. Authors should refrain from making bold conclusions based on association analysis of TEs and genes and/or biological pathways unless the association is already

published with a robust, reliable and reproducible set of experiments. A more thorough discussion identifying the gene sets or proposing targeted experiments would greatly strengthen the manuscript's premise.

3. Authors do not discuss or perform any analysis on an alternate hypothesis of how LTR5_HS chromatin can affect the cell migrations. While most of the LTR5_HS elements are solo in the human genome, a few of them encode full-length proviral sequences that express Rec, Env and Np9 subunits. It would be nice to see if the CRISPRi targeted LTR5_HS exist as solo LTRs or if they are promoters of HERVK proviruses. We and others have shown that these Rec and Env subunits can effectively influence cell migrations. I encourage authors to discuss for clarity. For instance, LTR5_HS-HERVK-Rec subunits control the cell migration of melanocytes in melanoma (PMID: 33202765), which has the same embryonic origin as CNCCs.

4. It is unclear that LTR5_HS or SVA insertions drive the formation of new open chromatin states. It is equally plausible that LTR5_HS elements preferentially integrate into preexisting open chromatin regions rather than creating novel enhancers. This does not affect the conclusions drawn in this study, although it will be important to interpret to what extent these elements drive evolutionary evolution. I suggest reanalysing Chimpanzee neural crest ATAC-seq signals in human-chimpanzee orthologous sequences of flanking regions of human LTR5_HS integrations. A more detailed discussion of these observations to distinguish between these scenarios would significantly strengthen the authors' conclusions.

5. Regarding target gene selection, it is crucial to distinguish whether a given enhancer functions as the sole regulatory element for its target gene or operates alongside multiple enhancers. Genes often possess multiple enhancers, including primary and shadow enhancers, which can act redundantly to ensure robust gene expression. Therefore, attributing gene regulation to a single enhancer without considering the broader enhancer landscape may lead to oversimplified conclusions. I recommend that the authors assess the enhancer redundancy within the genomic context of each target gene to accurately interpret the regulatory mechanisms involved.

6. Figure 4 displays the chromatin accessibility of various tissues, which are compared with cell lines. What is the rationale for selecting the shown tissues? How does the accessibility of the Lung and meso-endoderm relate to the story presented here? Chromatin accessibility is highly cell-type specific, which may not be seen in the tissues as they are a messy mix of so many cell types. There are plenty of ATAC-seq data publicly available on cell types in case the authors want to go down that road.

• Minor Comments:

1. There are inconsistencies in capitalising the texts, e.g., SINE-Vntr-Alus on page 1 of the Introduction should be SINE-VNTR-Alu, and capitalising the y-axis on figure 2 is distracting.
2. The authors used the phrases "decorated by the H3K27ac" and "in a study by the Trono group." I am sure they could have been written differently, such as "deposition/modification" and "seminal studies" or the name of the laboratory, respectively.
3. The way I see 1F is that TFAP2A is downregulated in CRISPRi CNCCs, although it is statistically insignificant at the protein level, but the trend is towards the decline in expression. I think this is a nice result as it can be associated with the inherent function of TFAP2A in facilitating the migration of cells. It is worth discussing in a single sentence.
4. Referring to the ChIP-seq experiments, the Authors state that "TEs are directly bound by TWIST1 in human CNCCs". The issue is that ChIP-seq doesn't tell us if the protein of interest directly binds to the sequenced fragments.
5. While discussing the chromatin profile of multiple organs in Figure 4 and elsewhere in the manuscript, the Authors declare the Modus operandi on these enhancers/ATAC-seq regions derived from LTR5_HS is "co-opted". The co-option is a strong statement based on the analysis presented here. Such substantial statements warrant further functional validation. Specifically, experiments such as ChIP-STARR-seq would be necessary to confirm the enhancer activity of these LTR5_HS elements, followed by genetic perturbation studies to demonstrate that disruption of these enhancers leads to dysregulation of target gene expression and impacts the associated phenotype. I do not recommend that the authors provide additional experimental evidence, but they should tone down the claim of co-option.

Bests

Manu

Reviewer #2:

In this manuscript by Deelen et al. entitled "Human-specific transposons rewired the gene regulatory networks essential for neural crest migration" the authors investigate the domestication of human-specific transposable elements as transcriptional enhancers during cranial neural crest cell (CNCC) specification. By using human iPSC-derived CNCCs they described how human specific TEs (SVA E/F and LTR5Hs retrotransposons) act as active enhancers. While the research questions regarding the enhancer activity of SVAs and LTR retrotransposons is very interesting and timely, there are several uncertainties in this manuscript, including problems with the experimental design, which makes it unsuitable for publication at this stage.

Major comments

1. Our main concern with this manuscript is that throughout the study, the authors consider SVA E/Fs and LTR5Hs collectively as human specific transposons. However, the mechanisms by which these very different transposons may influence gene regulatory networks is likely to be very different. Therefore, they cannot be considered under one "umbrella" of human specific transposons and the data should be analysed separately focusing on the two different families of TEs. However, since the guide RNAs that the authors used to silence transposon are not specific for any of the families, this is challenging (to say the least). Moreover, the guide RNAs target all SVAs, not only human-specific ones. Therefore, the conclusions that the enhancer effect comes from human-specific transposons also cannot be drawn.

2. The CRISPRi experiments need to be further characterised in relation to potential off-target effects as well in relation to the on-target effects. For example, there is no transcriptomic data showing the efficiency of the silencing. While looking at SVA expression in RNA-seq data is challenging, they should be able to study the silencing of the LTR5Hs transposons using RNA sequencing. For the SVA retrotransposons they could have used H3K4me3 Chip-seq or CUT&RUN sequencing to see if there is active transcription of the insertions.

3. The characterisation of the CNCC model is incomplete. How similar is this model in relation to the human situation. Further detailed characterisation of the model system would be essential to interpret the results. For example, how does the transposon expression landscape look in the different cell type?

4. To be able to say that the alterations in gene expression upon CRISPRi have a functional consequence, further functional assays are needed. As it is now the functional characterisation is limited to one experimental setup. Orthogonal approaches would be very valuable.

5. It is not clear why in Figure 3 the authors focus their GO analysis on genes that are differentially expressed genome wide and not only on the 107 genes that are nearby the accessible human-specific TEs. By using this approach, it is not possible to determine if the alterations in cell migration showed in figure 4 are due to the effect of transposable elements and therefore the main message of the paper is not supported by data.

6. The authors rely on a unique mapping approach throughout the manuscript. While this is most likely the best approach for the dataset, it also comes with limitation (in particular in regard to the analysis of SVAs). The authors should acknowledge these limitations and how it impacts on their conclusions.

In summary, although this is an interesting area of research the data presented in this manuscript is very difficult to interpret. We would recommend that the experiments are repeated using guide RNAs that are family and subfamily specific and don't target several families at the time. As it stands now, the results from this work cannot be interpreted and the main conclusions in the paper are not supported by the data.

Reviewer #3:

This manuscript by Deelen et al. explores the co-option of human-specific transposable elements (TEs) in the context of cranial neural crest cell (CNCC) specification. The authors propose that this species-specific regulatory phenomenon is more likely driven by changes in gene regulation rather than mutations in protein-coding sequences. They focus on two TE families-SVA-E/F and LTR5Hs-whose subfamily divergence coincides with the evolutionary split between humans and chimpanzees.

Using an inducible CRISPR interference (CRISPRi) system in human induced pluripotent stem cells (iPSCs), the authors targeted approximately 80% of all annotated SVAs and LTR5Hs in the human genome. Upon differentiation into the CNCC lineage and through an integrated analysis of ATAC-seq and RNA-seq data, they identified ~250 human-specific elements from these families that are highly accessible in CNCCs. Notably, repression of these elements leads to attenuated expression of CNCC-related genes, supporting the functional relevance of these TEs in human-specific developmental regulation.

This is a well-designed and clearly described study that offers novel insight into the regulatory contribution of TEs to human evolution. I have no reservations regarding its suitability for publication in *Molecular Systems Biology*. My comments are minor and primarily aimed at improving clarity:

Throughout the manuscript, the terminology alternates between "hominoid-specific" and "human-specific" LTR5Hs and SVAs, which can be confusing. Given the study's focus on human-specific regulatory innovation, it would be helpful to consistently distinguish these categories and de-emphasize the broader hominoid-specific elements unless directly relevant.

The authors state that 75% of LTR5Hs and SVA elements are effectively repressed by their CRISPRi system. Is there information available about the remaining 25%-specifically, which genes they are near or whether they exhibit any distinguishing features?

Regarding the statement: "In total, we identified 107 genes located near these elements that were actively expressed in CNCCs (median TPM >1 across all -gRNA replicates)," it would be helpful to clarify how this expression threshold was determined and whether other cutoffs were considered.

Reviewer #1:

This study from Marco Trizzino and colleagues finds an important and innovative evolutionary detail of cranial neural crest cell (CNCC) specification, which appears, in part, to be mediated by human-specific Endogenous Retroviruses (LTR5_Hs). These HERVs are the youngest ones in our genome that are remnants of ancient viruses that infiltrated the genomes of our ancestors after the human-chimp split (< 6 million years ago). Initially, these viral invaders might have behaved like deleterious insertions as evident from their major existence as solo-LTRs in our genome, which over time have become the special sites for regulating the expression of neighbor genes. Although a single LTR5_Hs might have a small effect, the cumulative influence of hundreds of them can dramatically impact how our genes respond during the cell specification, migration or even differentiation. The novel findings of this research are the discovery that these elements are not static relics; they have been continuously refined by evolution to select them as contributors to human-specific features. In particular, this research shows that LTR5_Hs have evolved precise TWIST1 binding motifs within the chromatin accessible regions of CNCC and demonstrate that LTR5Hs and SVAs probably function as enhancers within human CNCCs. Overlaying chromatin accessibility, transcriptomics and genetic perturbation using state-of-the-art CRISPR-interference, the authors suggest that around 500 of their loci are functional enhancers for 107 genes that plausibly lead to CNCC migration. A few of these genes are also among those whose expression is gained to be human-specific in CNCCs.

This research also answers, to some extent, why the youngest LTRs and SVAs underwent rapid expansion in the common ancestor post-split of humans and chimpanzees. This rapid expansion suggests that these elements provided ready-made genetic tools during periods of rapid environmental change, allowing our ancestors to fine-tune their cranial morphology swiftly and effectively. This adaptive mechanism, forged over millions of years, not only explains the rapid evolution of our craniofacial structures and peripheral nervous system but may also account for the genetic basis of susceptibility to related diseases in modern humans.

I like this study as it gets close to answering why there is a remarkable regulatory potential of the youngest endogenous retroelements in the human genome. It suggests that these elements may have undergone positive selection driven by their potent enhancer activity to push the neural crest cells that renovate cranial structures in conjunction with peripheral neuronal networks, thereby contributing significantly to the host's regulatory responses upon environmental challenges. This work indeed represents a significant advance over previously published studies. Thus, this line of advancement has enormous potential to outreach plenty of molecular system biology audience, genome researchers, chromatin and transposon audiences.

Overall, although preliminary, the results are interesting and worthy of publishing. At this point, however, a few issues arise that need further clarification and maybe analysis before I consider this study complete.

We thank the Reviewer for the nice words of appreciation of our study. We have made all the suggested modifications to the paper, which improved the manuscript significantly.

Major Concerns:

My major concern is that the manuscript does not adequately address its limitations.

We completely agree, and apologies for this. Now, the discussion section has an extensive “limitations of the study” paragraph addressing all the limitations underlined by the Reviewer (see point-by-point below) and more.

1. For example, the authors have not used activity-by-contact (ABC) maps to infer looping interactions between LTR5_HS-derived enhancers and their target gene promoters.

We completely agree that this is a limitation of the study, but unfortunately enhancer-promoter contact maps have not been generated for CNCCs. Hence, in the updated manuscript we are now discussing this point as a limitation of our study. Specifically, we wrote the following:

“Finally, our study has several limitations..... Additionally, we assigned genes to TE-derived enhancers based on genomic proximity, but enhancer-promoter contact maps would offer more precise enhancer-gene associations. Unfortunately, genome-wide contact maps are currently unavailable for CNCCs.”

2. Moreover, it is unclear which specific genes are implicated and how these genes contribute to the migration of CNCCs. Authors have performed a pathway analysis on the differentially expressed genes, which is a substandard method to decipher the real physiological relevance. The investigations presented here are exploratory. Authors should refrain from making bold conclusions based on association analysis of TEs and genes and/or biological pathways unless the association is already published with a robust, reliable and reproducible set of experiments. A more thorough discussion identifying the gene sets or proposing targeted experiments would greatly strengthen the manuscript's premise.

Thanks for pointing this out, and we acknowledge it as a limitation as well, although we would like to highlight that in the paper, we have now included two different and orthogonal cell migration assays (transwell assay and wound healing assay) both showing significantly impaired cell migration upon TE-repression. Nonetheless, as suggested by the Reviewer, in the updated manuscript we are addressing this limitation in the discussion. Specifically, we wrote:

“Furthermore, while our Gene Ontology analysis robustly highlighted a “cell migration” signature, functionally validated through transwell and wound healing assays, we could not definitively identify the exact genes responsible for the altered migratory phenotype observed upon repression of hominoid-specific TEs”.

3. Authors do not discuss or perform any analysis on an alternate hypothesis of how LTR5_HS chromatin can affect the cell migrations. While most of the LTR5_HS elements are solo in the human genome, a few of them encode full-length proviral sequences that express Rec, Env and Np9 subunits. It would be nice to see if the CRISPRi targeted LTR5_HS exist as solo LTRs or if they are promoters of HERVK proviruses. We and others have shown that these Rec and Env subunits can effectively influence cell migrations. I encourage authors to discuss for clarity. For instance, LTR5_HS-HERVK-Rec subunits control the cell migration of melanocytes in melanoma (PMID: 33202765), which has the same embryonic origin as NCCs.

We thank the Reviewer for highlighting this very important point. We are now discussing the alternative scenario in the discussion section of the updated manuscript, and citing the suggested paper too. Specifically, we added the following paragraph: *“Importantly, while most of the LTR5HS elements are present as “solo LTRs” in the human genome, a few of them, also targeted by our gRNAs, encode full-length proviral sequences that express Rec, Env and Np9 subunits. Previous studies have shown that the*

Rec and Env subunits can effectively influence cell migration, for instance in melanocytes in melanoma (Singh et al., 2020). Therefore, we cannot rule out that the impairment of CNCC migration is not only due to repression of the LTR5HS-derived enhancer activity, but also to the repression of the encoded proteins of the full-length proviral sequences.”

4. It is unclear that LTR5_HS or SVA insertions drive the formation of new open chromatin states. It is equally plausible that LTR5_HS elements preferentially integrate into preexisting open chromatin regions rather than creating novel enhancers. This does not affect the conclusions drawn in this study, although it will be important to interpret to what extent these elements drive evolutionary evolution. I suggest reanalysing Chimpanzee neural crest ATAC-seq signals in human-chimpanzee orthologous sequences of flanking regions of human LTR5_HS integrations. A more detailed discussion of these observations to distinguish between these scenarios would significantly strengthen the authors' conclusions.

This is a great point, and we have now performed the suggested analysis. Specifically, we have isolated the flanking regions 250 bp up- and downstream of the accessible human-specific TEs, identified the chimpanzee orthologous coordinates, and used previously generated paired-end long chimpanzee CNCC ATAC-seq reads (Prescott et al. 2015) to look for accessibility. Strikingly, none of these sites was accessible in chimpanzee CNCCs (see new Supplementary Fig. S3). This suggests that the human-specific TEs predominantly created new enhancers rather than transposing into existing ones. However, a limitation of this approach (which we discuss) is that the chimp ATAC-seq was generated by a different laboratory so these findings should be taken with caution.

5. Regarding target gene selection, it is crucial to distinguish whether a given enhancer functions as the sole regulatory element for its target gene or operates alongside multiple enhancers. Genes often possess multiple enhancers, including primary and shadow enhancers, which can act redundantly to ensure robust gene expression. Therefore, attributing gene regulation to a single enhancer without considering the broader enhancer landscape may lead to oversimplified conclusions. I recommend that the authors assess the enhancer redundancy within the genomic context of each target gene to accurately interpret the regulatory mechanisms involved.

We agree that this is also a limitation, and in the absence of enhancer-promoter contact maps we are not really able to accurately assess redundancy. Therefore, we are also now discussing this as a limitation. Specifically, we wrote: *“Relatedly, the expression of a single gene is often regulated by multiple redundant enhancers, complicating the precise determination of individual enhancer contributions without CNCC-specific enhancer-promoter contact data. Consequently, the absence of a detectable gene expression change upon CRISPRi does not necessarily exclude enhancer activity of a specific TE, as compensatory enhancers might mask the loss-of-function phenotype”.*

6. Figure 5 displays the chromatin accessibility of various tissues, which are compared with cell lines. What is the rationale for selecting the shown tissues? How does the accessibility of the Lung and meso-endoderm relate to the story presented here? Chromatin accessibility is highly cell-type specific, which may not be seen in the tissues as they are a messy mix of so many cell types. There are plenty of ATAC-seq data publicly available on cell types in case the authors want to go down that road.

The aim of this analysis was to obtain a preliminary assessment of whether transposable elements (LTR5Hs and SVAs) that are accessible and potentially function as enhancers in cranial neural crest cells (CNCCs) also serve regulatory roles in other cell types.

To address this question, we utilized publicly available H3K27ac data from the Roadmap Epigenomics Project, encompassing 14 different cell types. While these data do not represent the full spectrum of human cell types, this preliminary analysis provided initial insights into the breadth of LTR5HS and SVA (potential) co-option as regulatory elements across lineages. Regarding the specific patterns observed in lung and meso-endoderm tissues, the meso-endoderm signal is biologically plausible, as TBR2, which is a transcription factor known to bind SVA elements (Patoori et al. 2022, Development), plays a key regulatory role in this lineage. This observation further supports the hypothesis that the availability of transcription factor binding sites within TEs is a critical driver of their regulatory co-option. In contrast, the signal observed in lung tissue appears noisy and may not represent genuine regulatory activity. Therefore, we would rather avoid overinterpreting this finding. We have incorporated a clarifying statement regarding these observations in the Results section.

Minor_Comments:

1. There are inconsistencies in capitalising the texts, e.g., SINE-Vntr-Alus on page 1 of the Introduction should be SINE-VNTR-Alu, and capitalising the y-axis on figure 2 is distracting. Fixed, thank you.
2. The authors used the phrases "decorated by the H3K27ac" and "in a study by the Trono group." I am sure they could have been written differently, such as "deposition/modification" and "seminal studies" or the name of the laboratory, respectively. Fixed, thank you.
3. The way I see 1F is that TFAP2A is downregulated in CRISPRi CNCCs, although it is statistically insignificant at the protein level, but the trend is towards the decline in expression. I think this is a nice result as it can be associated with the inherent function of TFAP2A in facilitating the migration of cells. It is worth discussing in a single sentence. Thank you, we are now highlighting this modest downregulation, and suggesting it might have some role in the observed phenotype.
4. Referring to the ChIP-seq experiments, the Authors state that "TEs are directly bound by TWIST1 in human CNCCs". The issue is that ChIP-seq doesn't tell us if the protein of interest directly binds to the sequenced fragments. Fixed, thank you.
5. While discussing the chromatin profile of multiple organs in Figure 4 and elsewhere in the manuscript, the Authors declare the Modus operandi on these enhancers/ATAC-seq regions derived from LTR5_HS is "co-opted". The co-option is a strong statement based on the analysis presented here. Such substantial statements warrant further functional validation. Specifically, experiments such as ChIP-STARR-seq would be necessary to confirm the enhancer activity of these LTR5_HS elements, followed by genetic perturbation studies to demonstrate that disruption of these enhancers leads to dysregulation of target gene expression and impacts the associated phenotype. I do not recommend that the authors provide additional experimental evidence, but they should tone down the claim of co-option. We have rephrased accordingly and also suggested in the discussion the validation of "co-option", by means of ChIP-STARR-seq and related approaches, would be needed to further support the mode.

Reviewer #2:

In this manuscript by Deelen et al. entitled "Human-specific transposons rewired the gene regulatory networks essential for neural crest migration" the authors investigate the domestication of human-specific transposable elements as transcriptional enhancers during cranial neural crest cell (CNCC) specification. By using human iPSC-derived CNCCs they described how human specific TEs (SVA E/F and LTR5Hs retrotransposons) act as active enhancers. While the research questions regarding the enhancer activity of SVAs and LTR retrotransposons is very interesting and timely, there are several uncertainties in this manuscript, including problems with the experimental design, which makes it unsuitable for publication at this stage.

We thank the Reviewer for the nice words of appreciation of our study. We agree with the criticisms, and we have made a significant effort to revise the manuscript extensively, aiming at addressing the issues identified by the Reviewer.

Major comments

1. Our main concern with this manuscript is that throughout the study, the authors consider SVA E/Fs and LTR5Hs collectively as human specific transposons. However, the mechanisms by which these very different transposons may influence gene regulatory networks is likely to be very different. Therefore, they cannot be considered under one "umbrella" of human specific transposons, and the data should be analysed separately focusing on the two different families of TEs. However, since the guide RNAs that the authors used to silence transposon are not specific for any of the families, this is challenging (to say the least). Moreover, the guide RNAs target all SVAs, not only human-specific ones. Therefore, the conclusions that the enhancer effect comes from human-specific transposons also cannot be drawn.

We agree with the Reviewer that since we used gRNAs targeting SVAs found in other hominoid species (i.e., other apes), we cannot claim that the observed phenotypes are caused by human-specific TEs. However, re-performing all experiments using gRNAs specific for LTR5Hs or individual SVA families (separating human-specific ones) was unfortunately not feasible for us. Therefore, following discussion with the Editor, we have made significant changes to the manuscript's interpretation and focus, beginning with the title and extending through the analysis, results interpretation, and discussion. Specifically, the revised manuscript now focuses on two families of young hominoid-specific TEs (LTR5Hs and SVAs) rather than human-specific TEs. Accordingly, the updated title is "**Hominoid-specific transposable elements rewired the expression of neural crest migration genes**". We also encourage the reviewer to suggest alternative titles that better fit with the approach and the key findings.

This reorganization applies throughout the results and discussion sections.

Moreover, as the Reviewer suggested, we now present separate analyses and results for SVAs and LTR5Hs, as well as combined analyses. This revision proved highly valuable, revealing that the majority of the "CNCC-specific" signal derives from LTR5Hs, with minimal contribution from SVAs. For example, by separating LTR5Hs from SVAs in our analysis, we discovered that LTR5Hs sequences show very strong enrichment for binding motifs of CNCC-signature transcription factors

(e.g., TWIST1, SOX9/10), with >80% of accessible LTR5Hs containing these motifs, while SVAs show much weaker enrichment for the same motifs (see new Supplementary Fig. S2). This may explain why the majority of LTR5Hs are accessible in human CNCCs, while only a small minority of all existing SVAs (and **specifically only 5% of all SVAs**) are accessible in the same cell type.

Consistent with this finding, re-analysis of TWIST1 ChIP-seq data revealed that the vast majority (77%) of TEs bound by TWIST1 in CNCCs (see Fig. 2e) are LTR5Hs. Similarly, when examining the 107 nearest expressed genes near accessible LTR5Hs or SVAs, we found that the overwhelming majority of genes losing expression upon CRISPR targeting are located near LTR5Hs rather than SVAs (71 of 83 genes losing expression, 86%).

In summary, we thank the Reviewer for suggesting this breakdown by TE family, as the results were highly insightful, indicating that LTR5Hs have a seemingly more important role than SVAs in CNCC specification. We would also note that the gRNA set used to target SVAs and LTR5Hs has been employed in multiple studies by several groups, including Didier Trono's group, which designed and first used them in their seminal 2019 Cell Stem Cell paper (Pontis et al. 2019).

2. The CRISPRi experiments need to be further characterised in relation to potential off-target effects as well in relation to the on-target effects. For example, there is no transcriptomic data showing the efficiency of the silencing. While looking at SVA expression in RNA-seq data is challenging, they should be able to study the silencing of the LTR5Hs transposons using RNA sequencing. For the SVA retrotransposons they could have used H3K4me3 Chip-seq or CUT&RUN sequencing to see if there is active transcription of the insertions.

We thank the Reviewer for suggesting this additional analysis. We initially attempted using our RNA-seq data but, as the Reviewer noted, the short reads combined with the young age of the investigated TE family did not allow sufficient signal resolution for detection at SVAs and LTR5Hs. Therefore, following the Reviewer's suggestion, we performed H3K4me3 ChIP-seq with long paired-end reads, which provided strong supporting evidence. Specifically, the H3K4me3 analysis revealed that approximately 40% of the LTR5Hs and SVAs detected as accessible in CNCCs were also decorated by H3K4me3. Importantly, H3K4me3 signal was significantly reduced upon CRISPR-i induction at all these elements (Supplementary Fig. S1C). Overall, these additional experiments further supported that CRISPR-i is effective in repressing the LTR5Hs and SVAs in CNCCs. We note that our accessible LTR5Hs and SVAs are also decorated by another active histone modification, and specifically H3K27ac (see Supplementary Fig. S1b).

3. The characterisation of the CNCC model is incomplete. How similar is this model in relation to the human situation. Further detailed characterisation of the model system would be essential to interpret the results. For example, how does the transposon expression landscape look in the different cell type?

Regarding the first point, "*How similar is this model in relation to the human situation?*": In vitro modeling of cranial neural crest cell (CNCC) development using induced pluripotent stem cells (iPSCs) or embryonic stem cells (ESCs) has become an established approach in neural crest biology over the past 15 years (see, for example, PMIDs: 26839343, 20130577, 26365491, 22981823, 32991838, 34753942, 39226899, 36220078). These studies have consistently demonstrated that neural crest cells derived in vitro from iPSCs or ESCs exhibit high fidelity to

human neural crest cells across multiple parameters, including transcriptomic profiles, migratory behavior, and, most critically, their capacity to differentiate into diverse lineages such as craniofacial osteoblasts, chondrocytes, peripheral neurons, and melanocytes.

Additionally, we highlight that we employed two distinct differentiation protocols in parallel the present study (Supplementary Fig. 2C) and observed no significant protocol-dependent differences in the accessibility of LTR5Hs and SVAs, further validating the robustness of our model system. Finally, in response to the Reviewer's recommendation, the revised manuscript now includes a supplementary figure displaying the comprehensive expression landscape of all major transposable element families and subfamilies in our iPSC-derived CNCC model.

4. To be able to say that the alterations in gene expression upon CRISPRi have a functional consequence, further functional assays are needed. As it is now the functional characterisation is limited to one experimental setup. Orthogonal approaches would be valuable.

The Reviewer is absolutely correct on this point. In response to this recommendation, we performed a second functional assay, and specifically a wound-healing assay (scratch assay), which yielded results entirely consistent with our previously conducted transwell migration assay. Therefore, in the revised manuscript we now present two orthogonal functional assays that collectively demonstrate impaired CNCC migration following CRISPR interference. These findings are presented in the updated Figure 4, which is included below for the reviewer's convenience. We thank the reviewer for this valuable suggestion, as the incorporation of two independent validation assays significantly strengthens the manuscript and provides robust evidence for our conclusions.

Fig. 4d-e: scratch (wound ealing) assay results, confirming that cell migration is impaired upon CRISPR-I (+gRNAs) in neural crest cells.

5. It is not clear why in Figure 3 the authors focus their GO analysis on genes that are differentially expressed genome wide and not only on the 107 genes that are nearby the accessible human-specific TEs. By using this approach, it is not possible to determine if the alterations in cell migration showed in figure 4 are due to the effect of transposable elements and therefore the main message of the paper is not supported by data.

Following the Reviewer's suggestion, we conducted Gene Ontology (GO) term analysis on the 107 genes and found that cell migration was among the top five most significantly enriched pathways. This finding, combined with validation through two orthogonal functional assays, provides robust evidence that the transposable elements under investigation contribute to the regulation of CNCC migration.

6. The authors rely on a unique mapping approach throughout the manuscript. While this is most likely the best approach for the dataset, it also comes with limitation (in particular in regard to the analysis of SVAs). The authors should acknowledge these limitations and how it impacts on their conclusions.

We completely agree with the Reviewer and have added a paragraph in the discussion to acknowledge this.

Reviewer#3:

This manuscript by Deelen et al. explores the co-option of human-specific transposable elements (TEs) in the context of cranial neural crest cell (CNCC) specification. The authors propose that this species-specific regulatory phenomenon is more likely driven by changes in gene regulation rather than mutations in protein-coding sequences. They focus on two TE families-SVA-E/F and LTR5Hs-whose subfamily divergence coincides with the evolutionary split between humans and chimpanzees.

Using an inducible CRISPR interference (CRISPRi) system in human induced pluripotent stem cells (iPSCs), the authors targeted approximately 80% of all annotated SVAs and LTR5Hs in the human genome. Upon differentiation into the CNCC lineage and through an integrated analysis of ATAC-seq and RNA-seq data, they identified ~250 human-specific elements from these families that are highly accessible in CNCCs. Notably, repression of these elements leads to attenuated expression of CNCC-related genes, supporting the functional relevance of these TEs in human-specific developmental regulation. This is a well-designed and clearly described study that offers novel insight into the regulatory contribution of TEs to human evolution. I have no reservations regarding its suitability for publication in Molecular Systems Biology.

We are thankful to the Reviewer for the VERY positive words of appreciation of our manuscript.

My comments are minor and primarily aimed at improving clarity:

Throughout the manuscript, the terminology alternates between "hominoid-specific" and "human-specific" LTR5Hs and SVAs, which can be confusing. Given the study's focus on human-specific regulatory innovation, it would be helpful to consistently distinguish these categories and de-emphasize the broader hominoid-specific elements unless directly relevant.

We agree with this comment, and also in the light of Reviewer-2's concerns we have now switched all the related terminology in the manuscript to "hominoid-specific".

The authors state that 75% of LTR5Hs and SVA elements are effectively repressed by their CRISPRi system. Is there information available about the remaining 25%-

specifically, which genes they are near or whether they exhibit any distinguishing features?

The remaining 25% comprise a mixture of LTR5Hs and SVAs, exhibiting a distribution pattern nearly identical to the targeted 75% in terms of relative representation across different TE families. Motif analysis of this 25% subset failed to identify binding sites for key neural crest transcription factors, including TWIST1, SOX9/10, TFAP2A, or other known regulatory coordinators. Gene ontology analysis of genes proximal to these non-targeted TEs revealed no enrichment for pathways associated with neural crest specification or migration. These findings have been incorporated into the discussion section.

Regarding the statement: "In total, we identified 107 genes located near these elements that were actively expressed in CNCCs (median TPM >1 across all -gRNA replicates)," it would be helpful to clarify how this expression threshold was determined and whether other cutoffs were considered.

The Reviewer raises an important question. Using $\text{TPM} \geq 1$ as a cutoff for expressed genes is a very common heuristic approach in RNA-seq analyses, but it is not a hard rule. It is usually chosen to filter out noise from real signal. In fact, TPM values below 1 often reflect stochastic low-level transcription or technical noise, so setting a threshold at 1 helps focussing on genes with more robust expression.

Consistent with this, many differential-expression pipelines (DESeq2, edgeR, limma) recommend filtering out genes with consistently low counts, recommending $\text{TPM} \geq 1$. Again, it is not a hard-rule, but we felt it would be the most appropriate threshold to use given the experimental and analytical set up.

20th Aug 2025

Manuscript Number: MSB-2025-13036R

Title: Hominoid-specific transposable elements rewired the expression of neural crest migration genes

Author: Laura Deelen

Zoe Mitchell

Martina Demurtas

Andria Koulle

Beatriz Garcia Del Valle

MARCO TRIZZINO

Dear Dr. Trizzino,

Thank you again for submitting your work to Molecular Systems Biology. We have now heard back from the three referees who accepted to evaluate the study. As you will see, the referees are happy with your revision and recommend acceptance. Yet, one of them still has a minor issue they would like to address before final acceptance. Moreover, our editorial assistance team flagged a couple of more technical (yet very important) issues we would need you to fix. You will find all the relevant information copied below.

Please resubmit your revised manuscript online, with a covering letter listing amendments and responses to the last points raised by the referee (there is no need to formally address the points raised by our assistance team in the response letter). Please resubmit the paper ****within one month****. Please use the Manuscript Number (above) in all correspondence.

When you resubmit your manuscript, please download our CHECKLIST (<https://bit.ly/EMBOPressAuthorChecklist>) and include the completed form in your submission. **Please note** that the Author Checklist will be published alongside the paper as part of the transparent process (<https://www.embopress.org/page/journal/17444292/authorguide#transparentprocess>)

Click on the link below to submit your revised paper.

Thank you for submitting this excellent paper to Molecular Systems Biology!

Best wishes,

Yehu Moran

Academic Editor

Molecular Systems Biology

Please click on the link below to submit the revision online before 19th Sep 2025.

IMPORTANT: When you send your revision, we will require the following items:

1. the manuscript text in LaTeX, RTF or MS Word format
2. a letter with a detailed description of the changes made in response to the referees. Please specify clearly the exact places in the text (pages and paragraphs) where each change has been made in response to each specific comment given
3. three to four 'bullet points' highlighting the main findings of your study
4. a short 'blurb' text summarizing in two sentences the study (max. 250 characters)
5. a 'thumbnail image' (550px width and max 400px height, Illustrator, PowerPoint or jpeg format), which can be used as 'visual title' for the synopsis section of your paper.
6. Please include an author contributions statement after the Acknowledgements section (see

<https://www.embopress.org/page/journal/17444292/authorguide#manuscriptpreparation>)

7. Please complete the CHECKLIST available at (<https://bit.ly/EMBOPressAuthorChecklist>). Please note that the Author Checklist will be published alongside the paper as part of the transparent process (<https://www.embopress.org/page/journal/17444292/authorguide#transparentprocess>).

See also figure legend guidelines: <https://www.embopress.org/page/journal/17444292/authorguide#figureformat>

9. Please note that corresponding authors are required to supply an ORCID ID for their name upon submission of a revised manuscript (EMBO Press signed a joint statement to encourage ORCID adoption).

(<https://www.embopress.org/page/journal/17444292/authorguide#editorialprocess>)

Currently, our records indicate that the ORCID for your account is 0000-0002-1383-7200.

Link Not Available

10. Include a Reagents and Tools Table as part of the Methods section, which can be downloaded from our author guidelines (<https://www.embopress.org/page/journal/17444292/authorguide#structuredmethods>)

*** PLEASE NOTE *** As part of the EMBO Press transparent editorial process initiative (see our Editorial at <https://dx.doi.org/10.1038/msb.2010.72> , Molecular Systems Biology will publish online a Review Process File to accompany accepted manuscripts. When preparing your letter of response, please be aware that in the event of acceptance, your cover letter/point-by-point document will be included as part of this File, which will be available to the scientific community. More information about this initiative is available in our Instructions to Authors. If you have any questions about this initiative, please contact the editorial office (msb@embo.org).

**** Specific comments by Editorial Assistance Team****

Data availability statement: included, but needs to go before Acknowledgments. Please fix.

AC/CRedit: needs to be removed from the manuscript text and appear only in the system.

REFERENCES: Problematic - et al needs to be used after 10 author names; DOIs should only be used for preprints and datasets that have not been published yet. Please correct.

FUNDING INFO: not congruent between the system and the manuscript text; missing in the system: grants BB/L015129/1 and BB/Y000854/1 from BBSRC, Wellcome Trust (grant 104931/Z/14/Z); on the other hand, if these are not supposed to be entered as funders in eJP (if the authors did not receive direct funding), then BB/Y000854/1 is the only item that needs to be in the system. Please correct.

APPENDIX FILE WITH Table of contents: included, but it needs a title page that has a table of contents with page numbers to show where each figure is located in the file; Appendix file should not contain huge datasets (Appendix File S1, Appendix File S2, etc.) and should not be over 400 pages long; each dataset should be uploaded as a separate Excel file using the following nomenclature Dataset EV1-EV8 (instead of Appendix File S1-S8).

SYNOPSIS IMAGE: included, but too large; it needs to be 550 pixels wide x 200-600 pixels high. Please correct.

SOURCE DATA: SD uploaded with completed checklist, some of the panels are deposited online, probably at GSE292478? - please check and verify.

NOTES:

- Materials and Methods should be renamed Methods

During our routine image checks, we noticed that at least some of the microscopy panels across the figure set appear pixelated (e.g., Fig. 1E). This is a common result of converting original 16-bit TIFF images to RGB format for publication, and while not a cause for concern, it can sometimes give the impression of image alteration to critical readers.

To avoid any such misunderstanding and to meet EMBO Press standards, we kindly ask that you Resubmit the complete figure set at the captured original data resolution.

Figure Legends - Comments

- Please indicate the statistical test used for data analysis in the legends of figures 2A, B; S4 B

- Please note that scale bar and its definition are missing for figure 1E. Please correct.

Data Citation - Comments

- Please note that the data callouts in the text for Kim et al., 2024), data citation does not include "Data ref.:" as a prefix.

****Specific comments by Reviewers****

Reviewer #1:

I've carefully reviewed both the updated figures/text and the point-by-point responses to the other reviewers' concerns, and I'm pleased to see that each of the issues raised has been addressed thoroughly and thoughtfully.

On balance, the revised manuscript now provides a coherent, compelling narrative and solid experimental support for the role of human/hominoid-specific ERV enhancers, mainly driven by LTR5_HS in cranial neural crest cell specification and migration. I have no further substantive concerns and am happy to recommend this manuscript for publication as revised.

I look forward to seeing it in print.

Sincerely,
Manu

Reviewer #2:

The authors have addressed all our comments and therefore the manuscript is now greatly improved. I have one minor comment to this version. The results of this manuscript show that it is mostly LTR5HS that are enriched for the binding sites but in the introduction and throughout the whole paper the focus is more on SVAs. It would be great if the authors could do a more detailed introduction to LTR5HS and try to rephrase the paper so it is clear that its focus is on LTR5HS and not SVAs.

Reviewer #3:

The authors have addressed all the raised critiques adequately so I would recommend this manuscript for publication.

All editorial and formatting issues were resolved by the authors.

31st Aug 2025

Manuscript number: MSB-2025-13036RR

Title: Hominoid-specific transposable elements reshaped neural crest migration in craniofacial development

Dear Dr. Trizzino,

Thank you again for sending us your revised manuscript. We are now satisfied with the modifications made and I am pleased to inform you that your paper has been accepted for publication. Thank you for for sending us this interesting study!

Yours sincerely,

Yehu Moran
Academic Editor
Molecular Systems Biology
